# VLM2Vec: Training Vision-Language Models for Massive Multimodal Embedding Tasks

**Ziyan Jiang**[1*], **Rui Meng**[2], **Xinyi Yang**[2], **Semih Yavuz**[2], **Yingbo Zhou**[2], **Wenhu Chen**[1]
[1]University of Waterloo, [2]Salesforce Research
ziyanjiang528@gmail.com, ruimeng@salesforce.com, wenhuchen@uwaterloo.ca

https://tiger-ai-lab.github.io/VLM2Vec/

## Abstract

Embedding models play a crucial role in a variety of downstream tasks, including semantic similarity, information retrieval, and clustering. While there has been a surge of interest in developing universal text embedding models that generalize across tasks (e.g., MTEB), progress in learning universal multimodal embedding models has been comparatively slow, despite their importance and practical applications. In this work, we explore the potential of building universal multimodal embeddings capable of handling a broad range of downstream tasks. Our contributions are two fold: (1) we propose MMEB (Massive Multimodal Embedding Benchmark), which covers 4 meta-tasks (i.e. classification, visual question answering, multimodal retrieval, and visual grounding) and 36 datasets, including 20 training datasets and 16 evaluation datasets covering both in-distribution and out-of-distribution tasks, and (2) VLM2Vec (Vision-Language Model → Vector), a contrastive training framework that converts any vision-language model into an embedding model via contrastive training on MMEB. Unlike previous models such as CLIP and BLIP, which encode text and images independently without task-specific guidance, VLM2Vec can process any combination of images and text while incorporating task instructions to generate a fixed-dimensional vector. We develop a series of VLM2Vec models based on state-of-the-art VLMs, including Phi-3.5-V, LLaVA-1.6, and Qwen2-VL, and evaluate them on MMEB's benchmark. With LoRA tuning, VLM2Vec achieves a 10% to 20% improvement over existing multimodal embedding models on MMEB's evaluation sets. Our findings reveal that VLMs are secretly strong embedding models.

## 1 Introduction

Embeddings, or distributed representations, encode inputs (whether text or images) as fixed-dimensional vectors, enabling a range of downstream tasks. Since the advent of Word2Vec (Mikolov, 2013) and GloVe (Pennington et al., 2014), substantial research efforts have focused on learning textual embeddings (Kiros et al., 2015; Conneau et al., 2017) and image embeddings (Radford et al., 2021; Li et al., 2022; Jia et al., 2021; Yu et al., 2022). These embeddings facilitate a variety of applications, including textual and visual semantic similarity (Agirre et al., 2012; Marelli et al., 2014; Chechik et al., 2010; Cer et al., 2017), information retrieval (Mitra et al., 2017; Karpukhin et al., 2020; Lin et al., 2014), automatic evaluation (Zhang et al., 2020; Sellam et al., 2020), prompt retrieval for in-context learning (Liu et al., 2022; Rubin et al., 2022; Hongjin et al., 2022), and retrieval-augmented generation (Lewis et al., 2020; Guu et al., 2020; Izacard & Grave, 2020). A recent shift in research has focused on developing universal embeddings that can generalize across a wide range of tasks. For instance, Muennighoff et al. (2023) introduced MTEB (Massive Text Embedding Benchmark) to comprehensively assess text embeddings across tasks such as classification and clustering. MTEB has become the standard for evaluating universal text embeddings. Recent works (Wang et al., 2022a; Su et al., 2023; Wang et al., 2024a; Springer et al., 2024; BehnamGhader et al., 2024) have demonstrated promising results on the MTEB benchmark. However, progress in multimodal embeddings has been relatively slower. Despite advancements

---

*Work done during an internship at University of Waterloo in collaboration with Salesforce Research. Corresponding authors are Ziyan Jiang, Rui Meng and Wenhu Chen

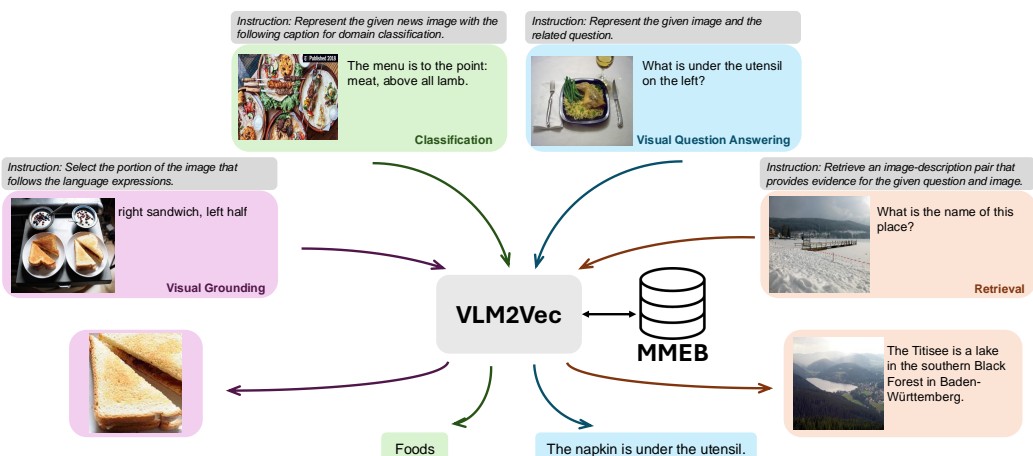

Figure 1: We develop a universal multimodal embedding benchmark, MMEB, along with VLM2VEC, an embedding model adapted from vision-language models (VLMs). VLM2VEC is capable of following instructions and performing various multimodal embedding tasks, accommodating any combination of image and text modalities.

in text embeddings, the lack of both benchmarks and methodologies in the multimodal embedding domain remains a challenge.

Current research in multimodal embeddings faces two primary limitations: (1) existing studies typically evaluate visual embeddings on isolated tasks, such as ImageNet classification (Deng et al., 2009; Hendrycks et al., 2021a;b) or MSCOCO/Flickr retrieval (Lin et al., 2014; Plummer et al., 2015); (2) most existing models, such as CLIP (Radford et al., 2021), BLIP (Li et al., 2022), and SigLIP (Zhai et al., 2023), either process text and images separately or perform shallow fusion of visual and textual information (Wei et al., 2023), limiting their ability to fully capture the relationships between text and image modalities. Furthermore, these models exhibit limited reasoning and generalization capabilities, particularly in zero-shot scenarios for complex reasoning tasks.

In this paper, we attempt to build an universal multimodal embedding framework to pave road for the future research, which consists of two efforts:

**- MMEB:** We introduce a novel benchmark, MMEB (Massive Multimodal Embedding Benchmark), which includes 36 datasets spanning four meta-task categories: classification, visual question answering, retrieval, and visual grounding. MMEB provides a comprehensive framework for training and evaluating embedding models across various combinations of text and image modalities. All tasks are reformulated as ranking tasks, where the model follows instructions, processes a query, and selects the correct target from a set of candidates. The query and target can be an image, text, or a combination of both. MMEB is divided into 20 in-distribution datasets, which can be used for training, and 16 out-of-distribution datasets, reserved for evaluation.

**- VLM2VEC:** We adopt the pre-trained vision-language models like Phi-3.5-V (Abdin et al., 2024) and LLaVA-1.6 (Li et al., 2024) as the backbone for VLM2VEC. In contrast to other multimodal embedding models like UniIR (Wei et al., 2023) and MagicLens (Zhang et al., 2024), which rely on late fusion of CLIP (Radford et al., 2021) features, our approach leverages the deep integration of vision and language features within a transformer architecture. There are several advantages to this approach: (1) VLMs are trained on massive multimodal datasets and can handle any combination of images and text, as well as high-resolution images and long text inputs; (2) vision and language features are deeply fused in the transformer model, improving the model's ability to capture cross-modal relationships; and (3) these models are well-suited for generalizing across diverse tasks, particularly those requiring instruction-following capabilities. These factors make VLM2VEC an ideal choice for task generalization. We trained VLM2VEC on the 20 MMEB training datasets using contrastive learning and compared its performance with various baselines.

Following extensive contrastive training, **VLM2VEC can handle any combination of images and text, producing fixed-dimensional vectors**. We evaluate VLM2VEC against a wide array of mul-

timodal embedding models, including CLIP (Radford et al., 2021), BLIP2 (Li et al., 2023a), SigLIP (Zhai et al., 2023), MagicLens (Zhang et al., 2024), UniIR (Wei et al., 2023) and E5-V (Jiang et al., 2024), demonstrating consistent improvements across all task categories. Notably, compared to the best baseline model without fine-tuning, our model achieves a 21.1 point improvement (from 44.7 to 65.8) across all 36 MMEB datasets and a 16.1-point increase (from 41.7 to 57.8) on 16 out-of-distribution datasets for zero-shot evaluation. Compared to the best baseline model with fine-tuning, our model achieves a 18.6 point improvement (from 47.2 to 65.8) across all 36 MMEB datasets and a 14.7-point increase (from 43.1 to 57.8) on 16 out-of-distribution datasets for zero-shot evaluation. Moreover, as a general multimodal representation model, VLM2VEC can still achieve competitive zero-shot T2I (Text-to-Image) and I2T (Image-to-Text) performance on Flickr30K compared to existing CLIP-like models, as presented in Table 11.

## 2 MMEB: A Benchmark for Multimodal Embeddings

### 2.1 Dataset Overview

We present MMEB (Massive Multimodal Embedding Benchmark), a comprehensive benchmark designed to evaluate multimodal embeddings across a diverse set of tasks. MMEB consists of 36 datasets organized into four meta-tasks: classification, visual question answering, retrieval, and visual grounding. Each task is reformulated as a ranking problem, where the model is provided with an instruction and a query (which may consist of text, images, or both) and is tasked with selecting the correct answer from a set of candidates. These candidates could be text, images, or additional instructions. The datasets are divided into two categories: 20 in-distribution datasets for training and 16 out-of-distribution datasets for evaluation. We report performance metrics across all 36 tasks. An overview of MMEB is provided in Figure 2 and the dataset statistics are provided in Table 1.

The embedding models are supposed to compress the query side into a vector and the target candidates into a set of vectors. The candidate with the highest dot-product will be selected as the prediction for evaluation. We measure the Precision@1 to reflect the percentage of top candidate matching the groundtruth. For the number of target candidates, a higher count could increase evaluation costs and hinder rapid model iteration, while a lower count might make the benchmark too simple and prone to saturation. To strike a balance between these extremes, we have chosen 1,000 candidates. Further details about this decision can be found in Section A.2.

MMEB offers a wide range of tasks from various domains, such as common, news, Wikipedia, web, and fashion. The benchmark incorporates diverse combinations of modalities for both queries and targets, including text, images, and text-image pairs. Additionally, tasks are designed to follow different types of instructions. For instance, tasks may involve object recognition (e.g., *"Identify the object shown in the image."*), retrieval (e.g., *"Find an image that matches the given caption."*), or visual grounding (e.g., *"Select the portion of the image that answers the question."*). Examples for each dataset in MMEB are provided in Tables 7, 8, 9 and 10. The diversity in MMEB makes it an ideal testbed for universal embeddings.

### 2.2 Meta-task and Dataset Design

MMEB is organized into four primary meta-task categories:

**Classification** The query consists of an instruction, an image, optionally accompanied by related text, while the target is the class label. The number of candidates equals the number of classes.

**Visual Question Answering** The query consists of an instruction, an image, and a piece of text as the question, while the target is the answer. Each query has 1 ground truth and 999 distractors as candidates.

**Information Retrieval** Both the query and target sides can involve a combination of text, images, and instructions. Each query has 1 ground truth and 999 distractors as candidates.

**Visual Grounding** The category is adapted from object detection tasks. The query combines an instruction (e.g., "Select the portion of the image that isolates the object of the given label: red apple") with the full image. This instruction guides the model to focus on a specific object within the image. Each candidate corresponds to cropped regions (bounding boxes) of the image, including both the object of interest and distractor regions. Each query includes 1,000 candidates: 1 ground

Table 1: The statistics of MMEB: 36 datasets across 4 meta-task categories, with 20 in-distribution datasets used for training and 16 out-of-distribution datasets used exclusively for evaluation.

| Meta-Task | Dataset | Query | Target | OOD? | #Training | #Eval | #Candidates |
|---|---|---|---|---|---|---|---|
| Classification (10 Tasks) | ImageNet-1K | I | T | | 100K | 1000 | 1000 |
| | N24News | I + T | I | | 49K | 1000 | 24 |
| | HatefulMemes | I | T | | 8K | 1000 | 2 |
| | VOC2007 | I | T | | 8K | 1000 | 20 |
| | SUN397 | I | T | | 20K | 1000 | 397 |
| | Place365 | I | T | ✓ | - | 1000 | 365 |
| | ImageNet-A | I | T | ✓ | - | 1000 | 1000 |
| | ImageNet-R | I | T | ✓ | - | 1000 | 200 |
| | ObjectNet | I | T | ✓ | - | 1000 | 313 |
| | Country-211 | I | T | ✓ | - | 1000 | 211 |
| VQA (10 Tasks) | OK-VQA | I + T | T | | 9K | 1000 | 1000 |
| | A-OKVQA | I + T | T | | 17K | 1000 | 1000 |
| | DocVQA | I + T | T | | 40K | 1000 | 1000 |
| | InfographicVQA | I + T | T | | 24K | 1000 | 1000 |
| | ChartQA | I + T | T | | 28K | 1000 | 1000 |
| | Visual7W | I + T | T | | 70K | 1000 | 1000 |
| | ScienceQA | I + T | T | ✓ | - | 1000 | 1000 |
| | VizWiz | I + T | T | ✓ | - | 1000 | 1000 |
| | GQA | I + T | T | ✓ | - | 1000 | 1000 |
| | TextVQA | I + T | T | ✓ | - | 1000 | 1000 |
| Retrieval (12 Tasks) | VisDial | T | I | | 123K | 1000 | 1000 |
| | CIRR | I + T | I | | 26K | 1000 | 1000 |
| | VisualNews-t2i | T | I | | 100K | 1000 | 1000 |
| | VisualNews-i2t | I | T | | 100K | 1000 | 1000 |
| | MSCOCO-t2i | T | I | | 100K | 1000 | 1000 |
| | MSCOCO-i2t | I | T | | 113K | 1000 | 1000 |
| | NIGHTS | I | I | | 16K | 1000 | 1000 |
| | WebQA | T | I + T | | 17K | 1000 | 1000 |
| | OVEN | I + T | I + T | ✓ | - | 1000 | 1000 |
| | FashionIQ | I + T | I | ✓ | - | 1000 | 1000 |
| | EDIS | T | I + T | ✓ | - | 1000 | 1000 |
| | Wiki-SS-NQ | T | I | ✓ | - | 1000 | 1000 |
| Visual Grounding (4 Tasks) | MSCOCO | I + T | I | | 100K | 1000 | 1000 |
| | Visual7W-Pointing | I + T | I | ✓ | - | 1000 | 1000 |
| | RefCOCO | I + T | I | ✓ | - | 1000 | 1000 |
| | RefCOCO-Matching | I + T | I + T | ✓ | - | 1000 | 1000 |

truth and 999 distractors. These distractors may include hard negatives from the same object class, other objects in the image, or random objects from different images.

Further details on dataset processing can be found in Section A.1.

## 3 VLM2VEC: TRANSFORMING LVMS TO EMBEDDERS

### 3.1 CONTRASTIVE TRAINING

We develop VLM2VEC, a contrastive training framework designed to convert any state-of-the-art vision-language model into an embedding model, as illustrated in Figure 3. A relevant query-target pair is denoted as $(q, t^+)$. Both $q$ and $t^+$ could be either single image, text or single image + text. We define $q : (q_t, q_i)$ and $t^+ : (t_t^+, t_i^+)$.

We then apply the instruction to the original query $q$ to generate a new one $q_{\text{inst}}$:

$$q_{\text{inst}} = [\texttt{IMAGE\_TOKEN}] \, \text{Instruct:} \, \{task\_definition\} \, \backslash n \, \text{Query:} \, \{q\} \tag{1}$$

where "$\{task\_definition\}$" is a placeholder for a one-sentence description of the embedding task. To enhance the embedding model's generalizability by better understanding instructions, we have crafted task-specific instructions, as shown in Tables 7, 8, 9 and 10.

Given a pretrained VLM, we feed query and target into it to obtain the query and target embeddings $(\mathbf{h}_{q_{\text{inst}}}, \mathbf{h}_{t^+})$ by taking the last layer vector representation of the last token. To train the embedding model, we adopt the standard InfoNCE loss $\mathcal{L}$ over the in-batch negatives and hard negatives:

$$\min \quad \mathcal{L} = -\log \frac{\phi(\mathbf{h}_{q_{\text{inst}}}, \mathbf{h}_{t^+})}{\phi(\mathbf{h}_{q_{\text{inst}}}, \mathbf{h}_{t^+}) + \sum_{t^- \in \mathbb{N}} \phi(\mathbf{h}_{q_{\text{inst}}}, \mathbf{h}_{t^-})} \tag{2}$$

## MMEB: Massive Multimodal Embedding Benchmark

**Classification**

| | | |
|---|---|---|
| VOC2007 | N24News | SUN397 |
| ImageNet-1K | | HatefulMemes |
| ObjectNet | Country211 | Place365 |
| ImageNet-A | | ImageNet-R |

**VQA**

| | | |
|---|---|---|
| OK-VQA | A-OKVQA | DocVQA |
| InfoVQA | ChartQA | Visual7W |
| ScienceQA | | GQA |
| TextVQA | | VizWiz |

**Retrieval**

| | | |
|---|---|---|
| MSCOCO-i2t | MSCOCO-t2i | VisDial | CIRR |
| VisualNews-i2t | VisualNews-t2i | NIGHTS | WebQA |
| Wiki-SS-NQ | FashionIQ | OVEN | EDIS |

**Visual Grounding**

| | |
|---|---|
| MSCOCO | RefCOCO |
| RefCOCO-Matching | Visual7W-Pointing |

Figure 2: An overview of the tasks and datasets in MMEB. MMEB includes four meta-tasks and 36 datasets: 20 in-distribution datasets used for training and 16 out-of-distribution datasets used exclusively for evaluation.

where $\mathbb{N}$ denotes the set of all negatives, and $\phi(\mathbf{h}_q, \mathbf{h}_t)$ is a function that computes the matching score between query $q$ and target $t$. In this paper, we adopt the temperature-scaled cosine similarity function as $\phi(\mathbf{h}_q, \mathbf{h}_t) = \exp(\frac{1}{\tau}\cos(\mathbf{h}_q, \mathbf{h}_t))$, where $\tau$ is a temperature hyper-parameter.

### 3.2 INCREASING BATCH SIZE THROUGH GRADCACHE

Since hard negatives are often difficult or ambiguous to collect for most multimodal datasets, using larger batch sizes becomes crucial. This increases the number of in-batch random negatives, which in turn helps improve the performance of the embedding model.

A bottleneck lies in the GPU memory that limits us from increasing the batch size and the number of in-batch random negatives during training, as each training instance may include one image (either from the query or target side) or multiple images (from both query and target sides), resulting in substantial memory consumption. We apply GradCache (Gao et al., 2021a), a gradient caching technique that decouples backpropagation between contrastive loss and the encoder, removing encoder backward pass data dependency along the batch dimension.

Mathematically, supposed we have a large batch of queries $\mathcal{Q}$, and we divide it into a set of sub-batches, each of which can fit into memory for gradient computation: $\mathcal{Q} = \{\hat{Q}_1, \hat{Q}_2, \dots\}$. There are two major steps: "Representation Gradient Computation and Caching" and "Sub-batch Gradient Accumulation". First, gradient tensors within each subbatch is calculated and stored: $\mathbf{u}_i = \frac{\partial \mathcal{L}}{\partial f(q_i)}, \quad \mathbf{v}_i = \frac{\partial \mathcal{L}}{\partial f(t_i)}$.

Then gradients are accumulated for encoder parameters across all sub-batches:

$$\frac{\partial \mathcal{L}}{\partial \Theta} = \sum_{\hat{Q}_j \in \mathbb{Q}} \sum_{q_i \in \hat{Q}_j} \mathbf{u}_i \frac{\partial f(q_i)}{\partial \Theta} + \sum_{\hat{T}_j \in \mathbb{T}} \sum_{t_i \in \hat{T}_j} \mathbf{v}_i \frac{\partial f(t_i)}{\partial \Theta} \tag{3}$$

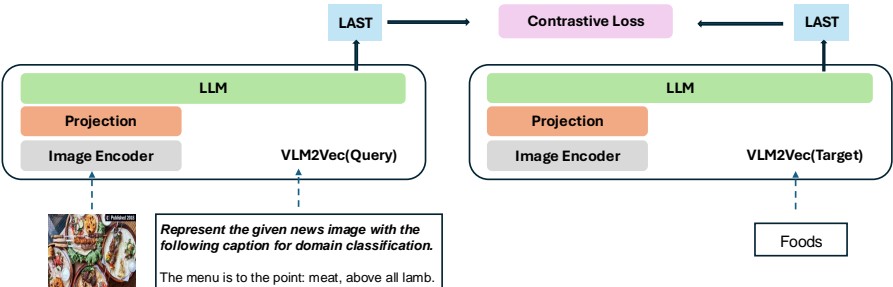

Figure 3: VLM2VEC uses a VLM as the backbone to deeply integrate image and text features. It is trained with a contrastive loss between the query and target, following task-specific instructions. The training data consists of diverse combinations of modalities on both the query and target sides, which may include images, text, or image-text pairs.

## 4 EXPERIMENTS

In this section, we adopt Phi-3.5-V (Abdin et al., 2024), LLaVA-1.6 (Li et al., 2024) and Qwen2-VL-7B-Instruct (Wang et al., 2024b) as the backbone VLMs, with training conducted via either full model fine-tuning or LoRA. The temperature for the loss function is set to 0.02, with a batch size of 1,024, a maximum text length of 256 tokens, and 2K training steps. The LoRA variant uses a rank of 8. For VLM2VEC leveraging Phi-3.5-V as the backbone, we configure the number of sub-image crops to 4. For VLM2VEC using LLaVA-1.6 and Qwen2-VL as the backbone, we resize the input images to a uniform resolution, employing two setups: a high-resolution configuration of 1344 × 1344 and a low-resolution configuration of 336 × 336. For the 20 training datasets, we randomly select up to 100K data points. When using GradCache, we set a sub-batch size accordingly, with the total batch size accumulated to 1,024. All experiments were run on 8 H100 GPUs. We report Precision@1 for all models in Table 2. It measures the ratio of positive candidates being ranked in the top place for all queries.

### 4.1 BASELINES

Four groups of baselines are reported in this study.

**CLIP-family:** We utilize vision/language encoders such as CLIP (Radford et al., 2021), Open-CLIP (Cherti et al., 2023), SigLIP (Zhai et al., 2023), and BLIP2 (Li et al., 2023a) as our baseline. Due to the length limitations of the text encoder, some queries or target text in certain tasks may be truncated. We apply score-level fusion by combining multimodal features using element-wise addition with equal weights ($w_1 = w_2 = 1$). We do not use instructions, as they could potentially degrade performance. For more details, please refer to Section 4.3.4.

**UniIR:** UniIR (Wei et al., 2023) is a unified, instruction-guided multimodal retriever designed to handle eight different retrieval tasks across multiple modalities. The model builds on CLIP and BLIP, employing shallow fusion techniques such as score-level and feature-level fusion to integrate modalities. In this study, we use the CLIP_SF and BLIP_FF variations as baselines.

**MagicLens:** MagicLens (Zhang et al., 2024) is a self-supervised image retrieval model capable of handling open-ended instructions. It utilizes a dual-encoder architecture with shared parameters, initializing the vision and language encoders with either CoCa or CLIP. The model uses a multi-head attention pooler to unify multimodal inputs into a single embedding. For this study, we report results using the CLIP-Large backbone.

**E5-V:** E5-V (Jiang et al., 2024) is a contemporary model that also leverages vision-language models for multimodal embedding tasks. It proposes a single-modality training approach, where the model is trained exclusively on text pairs. In contrast, our model is trained on multimodal pairs, which include various combinations of image and text modalities on both the query and target sides.

For all our baselines, we first use their original versions. Additionally, we have fine-tuned both CLIP and OpenCLIP on MMEB training datasets. We adopt the same experimental configurations

as VLM2VEC to ensure a fair comparison. For the remaining baseline models, UniIR and MagicLens also utilize a shallow fusion approach based on CLIP models, with their primary contribution being the datasets they were trained on. E5-V proposes training exclusively on text pairs, making it unsuitable for fine-tuning on our datasets. Therefore, we have not included the fine-tuned versions of these three models in this comparison.

## 4.2 MAIN RESULT

Table 2: Results on the MMEB benchmark. The scores are averaged per meta-task. For detailed scores per dataset, see Table 6. We include baselines with and without fine-tuning on MMEB training datasets and our models with three different backbones using a batch size of 1,024.

| Model | Per Meta-Task Score | | | | Average Score | | |
|---|---|---|---|---|---|---|---|
| | Classification | VQA | Retrieval | Grounding | IND | OOD | Overall |
| # of datasets → | 10 | 10 | 12 | 4 | 20 | 16 | 36 |
| *Baseline Models (No Fine-tuning on MMEB Training)* | | | | | | | |
| CLIP (Radford et al., 2021) | 42.8 | 9.1 | 53.0 | 51.8 | 37.1 | 38.7 | 37.8 |
| BLIP2 (Li et al., 2023a) | 27.0 | 4.2 | 33.9 | 47.0 | 25.3 | 25.1 | 25.2 |
| SigLIP (Zhai et al., 2023) | 40.3 | 8.4 | 31.6 | 59.5 | 32.3 | 38.0 | 34.8 |
| OpenCLIP (Cherti et al., 2023) | 47.8 | 10.9 | 52.3 | 53.3 | 39.3 | 40.2 | 39.7 |
| UniIR (BLIP_FF) (Wei et al., 2023) | 42.1 | 15.0 | 60.1 | 62.2 | 44.7 | 40.4 | 42.8 |
| UniIR (CLIP_SF) (Wei et al., 2023) | 44.3 | 16.2 | 61.8 | 65.3 | 47.1 | 41.7 | 44.7 |
| E5-V (Jiang et al., 2024) | 21.8 | 4.9 | 11.5 | 19.0 | 14.9 | 11.5 | 13.3 |
| Magiclens (Zhang et al., 2024) | 38.8 | 8.3 | 35.4 | 26.0 | 31.0 | 23.7 | 27.8 |
| *Baseline Models (Fine-tuning on MMEB Training)* | | | | | | | |
| CLIP | 55.2 | 19.7 | 53.2 | 62.2 | 47.6 | 42.8 | 45.4 |
| OpenCLIP | 56.0 | 21.9 | 55.4 | 64.1 | 50.5 | 43.1 | 47.2 |
| *Ours (VLM2VEC)* | | | | | | | |
| Phi-3.5-V, Full-model fine-tuned (#crop=4) | 52.8 | 50.3 | 57.8 | 72.3 | 62.8 | 47.4 | 55.9 |
| Phi-3.5-V, LoRA (#crop=4) | 54.8 | 54.9 | 62.3 | 79.5 | 66.5 | 52.0 | 60.1 |
| LLaVA-1.6, LoRA (low-res) | 54.7 | 50.3 | 56.2 | 64.0 | 61.0 | 47.5 | 55.0 |
| LLaVA-1.6, LoRA (high-res) | 61.2 | 49.9 | 67.4 | **86.1** | 67.5 | 57.1 | 62.9 |
| Qwen2-VL, LoRA (high-res) | **62.6** | **57.8** | **69.9** | 81.7 | **72.2** | **57.8** | **65.8** |
| Δ - Best baseline (No Fine-tuning) | +17.9 | +41.6 | +8.1 | +25.1 | +25.1 | +16.1 | +21.1 |
| Δ - Best baseline (Fine-tuning) | +6.6 | +35.9 | +14.5 | +22.0 | +21.7 | +14.7 | +18.6 |

From Table 2, the best variant of VLM2VEC leverages Qwen2-VL, is trained with LoRA, and processes input images at a relatively high resolution of 1344 × 1344. It achieves an average precision@1 of 65.8% across all 36 datasets from MMEB. Additionally, it maintains an average precision@1 of 57.8% on 16 out-of-distribution tasks in zero-shot evaluation, suggesting strong generalization ability. These results indicate that when trained on datasets spanning diverse task categories, domains, and modality combinations, VLM2VEC effectively follows instructions to align visual and textual representations while generalizing well to unseen tasks.

Furthermore, VLM2VEC using Phi-3.5-V and LLaVA-1.6 also outperforms baseline models. Notably, LLaVA-1.6 has a transparent pre-training data recipe with minimal overlap with MMEB's OOD datasets, reinforcing that the strong zero-shot performance of VLM2VEC is not due to prior exposure of its backbone to the OOD datasets. When using the same backbone, the full fine-tuning variant achieves slightly lower scores than the LoRA version. For a detailed discussion comparing full fine-tuning and LoRA, please refer to Section 4.3.1.

Compared to other baseline models, with or without fine-tuning on MMEB training data, our model demonstrates consistent improvements. Compared to the best baseline model without fine-tuning, our model achieves a 21.1 point improvement (from 44.7 to 65.8) across all 36 MMEB datasets and a 16.1-point increase (from 41.7 to 57.8) on 16 out-of-distribution datasets for zero-shot evaluation. Compared to the best baseline model with fine-tuning, our model achieves a 18.6 point improvement (from 47.2 to 65.8) across all 36 MMEB datasets and a 14.7-point increase (from 43.1 to 57.8) on 16 out-of-distribution datasets for zero-shot evaluation. Additionally, unlike the baseline models, which fail to demonstrate reasonable performance across all different task categories, VLM2VEC achieves strong performance across all four meta-task categories. This highlights its capability to handle a wide range of multimodal embedding tasks effectively.

## 4.3 RESULT ANALYSIS

To train an effective and generalizable multimodal embedding, various factors need to be considered, ranging from the data to the training setup. In this section, we present detailed ablation studies on these factors. We will discuss two training setups: Full Fine-Tuning vs. LoRA, along with Training parameters, and two topics related to data: Meta-task generalization and Impact of instructions.

### 4.3.1 FULL FINE-TUNING VS. LORA

When fine-tuning the VLMs, a key decision is whether to conduct full fine-tuning, which updates all parameters in the model, or to use a parameter-efficient method such as LoRA. We compare the performance of fully fine-tuned VLM2VEC with its LoRA variants at different ranks. The training and data setups are kept consistent across all models. We observe that LoRA achieves better performance when the rank is appropriately configured.

Table 3: We compare the performance of fully fine-tuned VLM2VEC with its LoRA variants at different ranks. LoRA can achieve better performance when the rank is appropriately configured. All the models utilize Phi-3.5-V as their backbone.

| Model | Meta-Task Average Score | | | | Average Score | | |
|---|---|---|---|---|---|---|---|
| | Classification | VQA | Retrieval | Grounding | IND | OOD | Overall |
| # of datasets → | 10 | 10 | 12 | 4 | 20 | 16 | 36 |
| Full Fine-Tuning (bs=256) | 50.4 | 46.4 | 52.6 | 68.6 | 57.9 | 44.7 | 52.0 |
| LoRA r = 4 (bs=256) | 52.7 | **53.6** | **60.1** | **80.2** | **64.9** | 50.4 | **58.4** |
| LoRA r = 8 (bs=256) | **52.9** | 52.5 | 60.3 | 80.0 | 64.2 | 50.8 | 58.2 |
| LoRA r = 16 (bs=256) | 51.1 | 40.5 | 52.0 | 72.5 | 54.9 | 45.8 | 50.8 |
| LoRA r = 32 (bs=256) | 50.6 | 47.8 | 53.9 | 72.5 | 58.9 | 46.5 | 53.4 |

### 4.3.2 TRAINING PARAMETERS

During our experiments, we identified three key parameters that significantly impact the performance of VLM2VEC: training batch size, the number of sub-image crops, and the number of training steps. In Figure 4, we observe that the final performance gradually improves as we increase the batch size, training step size, and number of sub-image crops. We particularly want to highlight the impact of batch size. Due to the lack of hard negatives, using a large batch size with plenty of random negatives, supported by the GradCache technique, plays a crucial role in enhancing the performance of VLM2VEC, as discussed in Section 3.2.

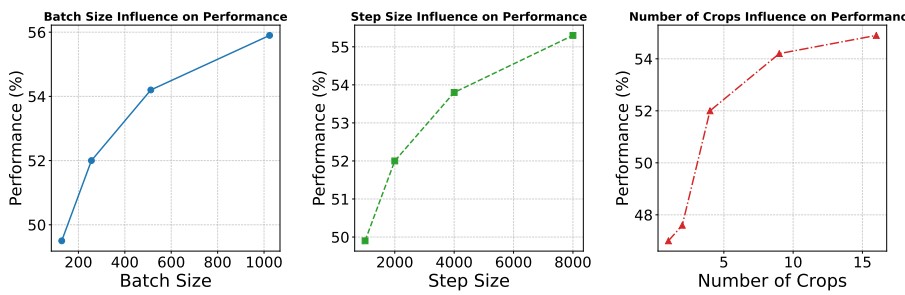

Figure 4: The figures demonstrate the influence of the training setup on VLM2VEC's final performance. Here, we examine the effects of training batch size, the number of sub-image crops, and the number of training steps. All the models utilize Phi-3.5-V as their backbone.

### 4.3.3 META-TASK GENERALIZATION

We have demonstrated that VLM2VEC has the potential to transfer to out-of-distribution datasets after being trained on a diverse range of in-distribution datasets, with the instruction-following set-

tings. An interesting question arises as to whether focusing on a specific meta-task can enhance the model's overall generalizability. We have trained three models, each focused solely on one meta-task (classification, visual question answering, and retrieval). Visual grounding was not included due to the limited number of training datasets. We then evaluated the models' transferability to other meta-tasks. We refer to these three models as VLM2VEC $_{\text{RET}}$, trained on 8 retrieval tasks, VLM2VEC $_{\text{VQA}}$, trained on 6 visual question answering tasks, and VLM2VEC $_{\text{CLS}}$, trained on 5 classification tasks.

Figure 5 illustrates the generalizability of these three models on unseen meta-tasks. We could observe that VLM2VEC $_{\text{RET}}$ has better generalizablilty on other meta-task, compared with other two models, especially on visual grounding categories. The reason is that retrieval tasks involve a more diverse combination of text and visual modalities from both the query and target sides, which helps the model generalize better to unseen meta-tasks. This observation highlights the benefits of using more diverse tasks in the VLM2VEC training process.

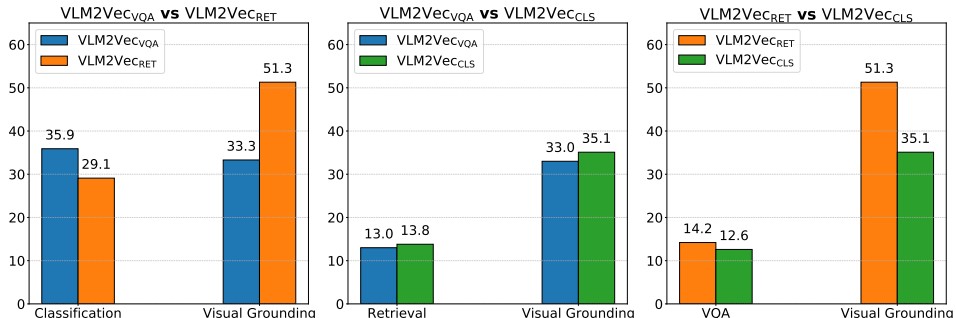

Figure 5: The figures show the generalization ability of models trained on one meta-task to other unseen meta-tasks. For example, the first subplot compares the performance of VLM2VEC trained exclusively on VQA tasks with VLM2VEC trained exclusively on retrieval tasks across the other two meta-task categories: classification and visual grounding. Overall, VLM2VEC trained on retrieval tasks demonstrate better generalization ability. All the models utilize Phi-3.5-V as their backbone.

### 4.3.4 IMPACT OF INSTRUCTIONS

Previous studies have shown the influence of instructions on addressing various tasks. VLM2VEC, which leverages a VLM as its backbone and is trained on large-scale datasets with instructions, is expected to better generalize across tasks and improve performance in multimodal embedding tasks. In this section, we evaluate the performance of both CLIP and VLM2VEC with and without task-specific instructions to quantify the impact of incorporating instructions into the embedding process. As shown in Table 4, incorporating instructions reduces the CLIP model's performance by 29.4%, while our VLM2VEC achieves a 49.4% improvement. This highlights our model's ability to follow instructions.

## 5 RELATED WORK

### 5.1 TEXT EMBEDDING

Text embeddings have demonstrated significant potential in powering downstream applications such as information retrieval (Karpukhin et al., 2020; Xiong et al., 2020), text similarity (Gao et al., 2021b), prompt retrieval for in-context learning (Hongjin et al., 2022), and classification (Logeswaran & Lee, 2018; Reimers & Gurevych, 2019). Early work focused on creating effective embeddings for specific tasks. With the rise of pretrained language models, efforts have shifted toward developing universal embedding models capable of handling a wide range of embedding tasks. Studies such as GTR (Ni et al., 2022) and E5 (Wang et al., 2022a) leveraged large amounts of noisy paired data to pretrain and fine-tune dense retrievers. More recent works like TART (Asai et al., 2022) and InstructOR (Su et al., 2023) introduced natural language prompts to guide embedding models in producing task-relevant embeddings. Building on this, models like E5Mistral(Wang et al., 2024a), SFR-Embedding(Meng et al., 2024), RepLLaMA(Ma et al., 2024b), GTE-Qwen2(Li et al., 2023b), and NV-Embed (Lee et al., 2024) have utilized pretrained large language models (LLMs)

Table 4: Comparison of CLIP and our VLM2VEC with and without task-specific instructions. Incorporating instructions could decrease CLIP's performance by 29.4%, whereas our VLM2VEC achieves a 49.4% improvement. VLM2VEC utilizes Phi-3.5-V as its backbone.

| Model | Meta-Task Average Score | | | | Average Score | | |
|---|---|---|---|---|---|---|---|
| | Classification | VQA | Retrieval | Grounding | IND | OOD | Overall |
| # of datasets → | 10 | 10 | 12 | 4 | 20 | 16 | 36 |
| *CILP* | | | | | | | |
| w/o instruction | 42.8 | 9.1 | 53.0 | 51.8 | 37.1 | 38.7 | 37.8 |
| w/ instruction | 17.4 | 8.0 | 41.3 | 52.9 | 23.8 | 30.3 | 26.7 |
| Δ | -59.3% | -12.1% | -22.1% | 2.1% | -35.8% | -21.7% | -29.4% |
| *Ours* (VLM2VEC) | | | | | | | |
| w/o instruction | 36.7 | 33.5 | 31.1 | 44.3 | 37.3 | 31.6 | 34.8 |
| w/ instruction | 50.4 | 46.4 | 52.6 | 68.6 | 57.9 | 44.7 | 52.0 |
| Δ | 37.3% | 38.5% | 69.1% | 54.9% | 55.2% | 41.5% | 49.4% |

as their backbone, fine-tuning them with multi-task data and instructions. These models have delivered significant improvements over earlier approaches that did not use LLMs for initialization or instruction tuning.

## 5.2 MULTIMODAL EMBEDDINGS

Multimodal embeddings have long been a significant research challenge. Early works like CLIP (Radford et al., 2021), BLIP (Li et al., 2022; 2023a), Align (Jia et al., 2021), SigLIP (Zhai et al., 2023), SimVLM (Wang et al., 2022b) and CoCa (Yu et al., 2022) primarily focused on learning universal representations from large-scale, weakly supervised image-text pairs. These models generally encode images and text separately, projecting them into a shared space. This approach has laid the groundwork for more recent multimodal models like LLaVA (Liu et al., 2024).

Most research on universal multimodal embeddings involves fine-tuning models like CLIP or BLIP, typically using simple fusion mechanisms to combine visual and language information. For instance, UniIR (Wei et al., 2023) creates multimodal embeddings by simply adding text and visual features, while MagicLens (Zhang et al., 2024) employs shallow self-attention layers to integrate these features more effectively. The study most similar to ours is E5-V (Jiang et al., 2024), a contemporary work that fine-tunes a vision-language model using only text training data.

## 5.3 EMBEDDING BENCHMARKS

Significant efforts have been made to develop benchmarks for evaluating retrieval systems. For text retrieval models, MS MARCO (Nguyen et al., 2016) and Natural Questions (Kwiatkowski et al., 2019b) are two of the most widely used benchmarks in general domains. To broaden the evaluation across more diverse domains, BEIR (Thakur et al.) was introduced, incorporating 18 datasets from various fields. Building on this, MTEB (Muennighoff et al., 2023) further expands BEIR's scope by adding more tasks, such as classification, clustering, and semantic textual similarity (STS).

For multimodal retrieval, several benchmarks have been introduced to evaluate model performance across different modalities. MBEIR (Wei et al., 2023) includes 8 tasks and 16 datasets, designed to test models' ability to retrieve information based on various forms of queries and instructions.

## 6 CONCLUSION

In this paper, we aim to build the first large-scale multimodal embedding framework, comprising two main components: MMEB and VLM2VEC. MMEB includes 36 datasets across four meta-task categories, providing a comprehensive and diverse framework for training and evaluating embedding models. VLM2VEC leverages VLMs as a backbone to deeply fuse visual and textual spaces, enhancing generalization to unseen tasks through instruction following.

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

# A    DETAILS OF MMEB

In this section, we provide additional details about our proposed benchmark, MMEB (Massive Multimodal Embedding Benchmark). Section A.1 outlines the specifics of the 36 datasets used in the MMEB benchmark. Section A.2 explains the process for determining the number of candidates in MMEB.

## A.1    DATASET DETAILS

### A.1.1    CLASSIFICATION

There are a total of 10 datasets for classification tasks.

**ImageNet-1K** (Deng et al., 2009) The dataset is s large-scale dataset commonly used in image classification, consisting of over 1 million images across 1K different classes.

**ImageNet-A** (Hendrycks et al., 2021b) The dataset contains images from a distribution unlike the ImageNet training distribution. ImageNet-A examples belong to ImageNet classes, but the examples are harder and can cause mistakes across various models. They cause consistent classification mistakes due to scene complications encountered in the long tail of scene configurations and by exploiting classifier blind spots.

**ImageNet-R** (Hendrycks et al., 2021a) The dataset contains set of images labeled with ImageNet labels obtained by collecting art, cartoons, deviantart, graffiti, embroidery, graphics, origami, paintings, patterns, plastic objects, plush objects, sculptures, sketches, tattoos, toys, and video game renditions of ImageNet classes.

**VOC2007** (Everingham et al., 2014) The dataset focuses on recognizing objects in realistic scenarios and contains 20 object classes.

**N24News** (Wang et al., 2021) The dataset is sourced from the New York Times and consists of 24 categories, with each news article containing both text and image information. The task is to classify the given news image and its accompanying text into one of the 24 categories.

**HatefulMemes** (Kiela et al., 2020) The dataset proposes a new challenge set for multimodal classification, focusing on detecting hate speech in multimodal memes.

**Place365** (Zhou et al., 2017) The dataset is a repository of 10 million scene photographs, labeled with scene semantic categories, comprising a large and diverse list of the types of environments encountered in the world.

**SUN397** (Xiao et al., 2010) The dataset is a dataset for scene recognition consisting of 397 categories.

**ObjectNet** (Barbu et al., 2019) The dataset is a crowd-sourced test set of 50K images featuring objects in unusual poses and cluttered scenes, designed to challenge recognition performance. It includes controls for rotation, background, and viewpoint, and covers 313 object classes.

**Country-211** (Radford et al., 2021) The dataset is designed to assess the geolocation capability of visual representations. It filters the YFCC100M dataset to find 211 countries that have at least 300 photos with GPS coordinates.

### A.1.2    VISUAL QUESTION ANSWERING (VQA)

There are a total of 10 datasets for VQA tasks.

**OK-VQA** (Marino et al., 2019) The dataset includes questions that require external resources for answers.

**A-OKVQA** (Schwenk et al., 2022) The dataset is an augmented successor of OK-VQA, requiring a broad base of commonsense and world knowledge to answer. The questions generally cannot be answered by simply querying a knowledge base, and instead require some form of commonsense reasoning about the scene depicted in the image.

**DocVQA** (Mathew et al., 2021) The dataset contains questions for document analysis and recognition over document images of various types and content.

**InfographicsVQA** (Mathew et al., 2022) The dataset comprises a diverse collection of infographics accompanied by natural language question and answer annotations. The questions require methods capable of jointly reasoning over the document layout, textual content, graphical elements, and data visualizations.

**ChartQA** (Masry et al., 2022) The dataset is designed for question answering about charts, with a focus on visual and logical reasoning applied to real-world charts.

**ScienceQA** (Lu et al., 2022) The dataset contains questions with diverse science topics and annotations of their answers with corresponding lectures and explanations.

**Visual7W-telling** (Zhu et al., 2016) The dataset establishes a semantic link between textual descriptions and image regions through object-level grounding. It has two types of questions: "telling" and "pointing". It leverages the six W questions (what, where, when, who, why, and how) to systematically examine a model's capability for visual understanding through telling questions. Additionally, a seventh "which" question is appended for visual answers as pointing questions. We use "Visual7W-telling" in our VQA category and "Visual7W-pointing" in our visual grounding category.

**VizWiz** (Gurari et al., 2018) The dataset originates from a natural visual question answering scenario, where blind individuals captured images and recorded spoken questions about them, along with 10 crowdsourced answers for each visual question. For our task, we select only the answerable questions.

**TextVQA** (Singh et al., 2019) The dataset is designed to benchmark visual reasoning based on text within images. Models need to read and reason about the text in images to answer related questions.

**GQA** (Hudson & Manning, 2019) The dataset is designed for real-world visual reasoning and compositional question answering. It uses real images from the Visual Genome dataset. Each image is accompanied by scene graph annotations that describe the classes and attributes of objects in the scene, as well as their pairwise relationships.

### A.1.3 RETRIEVAL

There are a total of 12 datasets for retrieval tasks.

**VisDial** (Das et al., 2017) The dataset features dialogues created by two Amazon Mechanical Turk workers. One worker takes the role of the "questioner", who only sees the text description of an image, while the other plays the "answerer", who has access to the image. They engage in a 10-round Q&A session about the image. We repurpose this dataset as a retrieval task, where the goal is to retrieve the image based on the given dialogue.

**CIRR** (Liu et al., 2021) The dataset is designed for the task of composed image retrieval. It consists of pairs of real-life reference and target images, along with a modification sentence that describes the changes made between the two images.

**FashionIQ** (Wu et al., 2021) The dataset contains images of fashion products with crowd-sourced descriptions highlighting the differences between these products. Similar to CIRR, FashionIQ can also be used for the task of composed image retrieval, where each test case consists of a pair of reference and target images, along with a modification sentence that describes the changes between the two images.

**VisualNews** (Liu et al., 2020) The dataset contains publicly available news image paired with captions. We split this task into two setups: **"VisualNews-i2t"**, which retrieves the caption given the news image and **"VisualNews-t2i"**, which retrieves the news image given the caption.

**MSCOCO** (Lin et al., 2014) The dataset is a well-known image caption dataset. Similar to VisualNews, WE split this task into two setups: **"MSCOCO-i2t"**, which retrieves the caption given the image and **"MSCOCO-t2i"**, which retrieves the image given the caption.

**WebQA** (Chang et al., 2022) The dataset is a multihop, multimodal QA dataset that requires retrieving a Wikipedia page to answer a given question. We use the Wikipedia page's image and text descriptions as the candidates for retrieval.

**NIGHTS** (Fu et al., 2023) The dataset contains human similarity judgments on image pairs that are alike in various ways. The original dataset consists of triplets: a reference image and two perturbed versions, along with human judgments indicating which version is most similar to the reference. Following M-BEIR (Wei et al., 2023), we refactor this dataset into a retrieval task to match pairwise images, where the reference image serves as the query, and the perturbed version that aligns with human judgment is the target.

**OVEN** (Hu et al., 2023) The dataset contains instances that include an image and a visual recognition text question. Additionally, each instance provides a related Wikipedia image along with its corresponding text description (the Wikipedia title and the first 100 tokens of its summary) as a reference for answering the question, which we treat as the target candidate.

**EDIS** (Liu et al., 2023) The dataset is a cross-modal image search in the news domain. This dataset contains entity-rich queries, requiring the model to understand both entities and events from the text queries. The candidate consists of the news image and its accompanying headline.

**Wiki-SS-NQ** (Ma et al., 2024a) The dataset is another retrieval-based VQA dataset. Unlike the original Natural Questions dataset (Kwiatkowski et al., 2019a), which uses a Wikipedia paragraph to answer the question, this dataset leverages Wiki-SS, utilizing Wikipedia page screenshots as the corpus. The screenshot provides more comprehensive information than a plain Wikipedia paragraph.

For **CIRR**, **FashionIQ**, **VisualNews**, **MSCOCO**, **WebQA**, **NIGHTS**, **OVEN** and **EDIS**, we use the processed versions from M-BEIR (Wei et al., 2023).

### A.1.4 VISUAL GROUNDING

There are a total of 4 datasets for visual grounding tasks.

**MSCOCO** (Lin et al., 2014) The dataset includes an object detection task, which involves recognizing an object from a given class in an image. We have repurposed this task into a ranking problem within the MMEB format. The query consists of the image and the object name, while the target is the cropped image of the specified object. We gather distractors from other objects in the same image as well as from different images. We discard test cases where the object is too small.

**RefCOCO** (Kazemzadeh et al., 2014) The dataset includes an object detection task that requires more reasoning than MSCOCO. Unlike simply identifying the object class, the RefCOCO dataset uses language expressions to refer to specific objects within an image. In our MMEB, we have two tasks related to RefCOCO: **"RefCOCO"** and **"RefCOCO-Matching"**. In "RefCOCO", the query consists of the image and the language expressions referring to a specific object, while the target is the cropped image of that object. In "RefCOCO-Matching", both the query and the target contain the image and the language expressions referring to a specific object, where the two objects are identical.

**Visual7W-pointing** (Zhu et al., 2016) The dataset establishes a semantic link between textual descriptions and image regions through object-level grounding. It has two types of questions: "telling" and "pointing". It leverages the six W questions (what, where, when, who, why, and how) to systematically examine a model's capability for visual understanding through telling questions. Additionally, a seventh "which" question is appended for visual answers as pointing questions. We use "Visual7W-telling" in our VQA category and "Visual7W-pointing" in our visual grounding category.

### A.2 SELECTION OF NUMBER OF CANDIDATES

A large number of candidates can make the benchmark more challenging and realistic. However, we also considered the computational cost when designing the benchmark. Choosing an excessively large number of candidates could result in very high inference costs, which may hinder rapid model iteration. As shown in Table 5, we compare the performance of VLM2VEC with different numbers of candidates in the MMEB benchmark. The results show that if the number of candidates is too small, the benchmark becomes saturated quickly. To balance evaluation cost with benchmark difficulty, we selected 1,000 as the optimal number of candidates.

Table 5: We compare the performance of VLM2VEC using different numbers of candidates in MMEB. To balance evaluation cost with benchmark difficulty, we selected 1,000 as the optimal number of candidates.

| #Candidates | Meta-Task Average Score | | | | Average Score | | |
|---|---|---|---|---|---|---|---|
| | Classification | VQA | Retrieval | Grounding | IND | OOD | Overall |
| # of datasets → | 10 | 10 | 12 | 4 | 20 | 16 | 36 |
| 100 | 54.8 | 81.8 | 86.1 | 89.6 | 85.2 | 65.9 | 76.6 |
| 500 | 54.8 | 65.9 | 72.6 | 82.8 | 74.6 | 57.3 | 66.9 |
| 1000 | 54.8 | 54.9 | 62.3 | 79.5 | 66.5 | 52.0 | 60.1 |
| 2000 | 54.8 | 50.1 | 56.7 | 71.0 | 62.2 | 48.0 | 55.9 |
| 5000 | 54.8 | 41.3 | 46.5 | 65.3 | 54.5 | 43.2 | 49.5 |

Table 6: The detailed results of the baselines and our VLM2VEC on MMEB, which includes 20 in-distribution datasets and 16 out-of-distribution datasets. The out-of-distribution datasets are highlighted with a yellow background in the table. VLM2VEC is built upon the LLaVA-1.6 backbone.

| | CLIP | OpenCLIP | SigLIP | BLIP2 | MagicLens | E5-V | UniIR | VLM2VEC |
|---|---|---|---|---|---|---|---|---|
| **Classification (10 tasks)** | | | | | | | | |
| ImageNet-1K | 55.8 | 63.5 | 45.4 | 10.3 | 48.0 | 9.6 | 58.3 | 74.5 |
| N24News | 34.7 | 38.6 | 13.9 | 36.0 | 33.7 | 23.4 | 42.5 | 80.3 |
| HatefulMemes | 51.1 | 51.7 | 47.2 | 49.6 | 49.0 | 49.7 | 56.4 | 67.9 |
| VOC2007 | 50.7 | 52.4 | 64.3 | 52.1 | 51.6 | 49.9 | 66.2 | 91.5 |
| SUN397 | 43.4 | 68.8 | 39.6 | 34.5 | 57.0 | 33.1 | 63.2 | 75.8 |
| Place365 | 28.5 | 37.8 | 20.0 | 21.5 | 31.5 | 8.6 | 36.5 | 44.0 |
| ImageNet-A | 25.5 | 14.2 | 42.6 | 3.2 | 8.0 | 2.0 | 9.8 | 43.6 |
| ImageNet-R | 75.6 | 83.0 | 75.0 | 39.7 | 70.9 | 30.8 | 66.2 | 79.8 |
| ObjectNet | 43.4 | 51.4 | 40.3 | 20.6 | 31.6 | 7.5 | 32.2 | 39.6 |
| Country-211 | 19.2 | 16.8 | 14.2 | 2.5 | 6.2 | 3.1 | 11.3 | 14.7 |
| *All Classification* | 42.8 | 47.8 | 40.3 | 27.0 | 38.8 | 21.8 | 44.3 | 61.2 |
| **VQA (10 tasks)** | | | | | | | | |
| OK-VQA | 7.5 | 11.5 | 2.4 | 8.7 | 12.7 | 8.9 | 25.4 | 69.0 |
| A-OKVQA | 3.8 | 3.3 | 1.5 | 3.2 | 2.9 | 5.9 | 8.8 | 54.4 |
| DocVQA | 4.0 | 5.3 | 4.2 | 2.6 | 3.0 | 1.7 | 6.2 | 52.0 |
| InfographicsVQA | 4.6 | 4.6 | 2.7 | 2.0 | 5.9 | 2.3 | 4.6 | 30.7 |
| ChartQA | 1.4 | 1.5 | 3.0 | 0.5 | 0.9 | 2.4 | 1.6 | 34.8 |
| Visual7W | 4.0 | 2.6 | 1.2 | 1.3 | 2.5 | 5.8 | 14.5 | 49.8 |
| ScienceQA | 9.4 | 10.2 | 7.9 | 6.8 | 5.2 | 3.6 | 12.8 | 42.1 |
| VizWiz | 8.2 | 6.6 | 2.3 | 4.0 | 1.7 | 2.6 | 24.3 | 43.0 |
| GQA | 41.3 | 52.5 | 57.5 | 9.7 | 43.5 | 7.8 | 48.8 | 61.2 |
| TextVQA | 7.0 | 10.9 | 1.0 | 3.3 | 4.6 | 8.2 | 15.1 | 62.0 |
| *All VQA* | 9.1 | 10.9 | 8.4 | 4.2 | 8.3 | 4.9 | 16.2 | 49.9 |
| **Retrieval (12 tasks)** | | | | | | | | |
| VisDial | 30.7 | 25.4 | 21.5 | 18.0 | 24.8 | 9.2 | 42.2 | 80.9 |
| CIRR | 12.6 | 15.4 | 15.1 | 9.8 | 39.1 | 6.1 | 51.3 | 49.9 |
| VisualNews-t2i | 78.9 | 74.0 | 51.0 | 48.1 | 50.7 | 13.5 | 74.3 | 75.4 |
| VisualNews-i2t | 79.6 | 78.0 | 52.4 | 13.5 | 21.1 | 8.1 | 76.8 | 80.0 |
| MSCOCO-t2i | 59.5 | 63.6 | 58.3 | 53.7 | 54.1 | 20.7 | 68.5 | 75.7 |
| MSCOCO-i2t | 57.7 | 62.1 | 55.0 | 20.3 | 40.0 | 14.0 | 72.1 | 73.1 |
| NIGHTS | 60.4 | 66.1 | 62.9 | 56.5 | 58.1 | 4.2 | 66.2 | 65.5 |
| WebQA | 67.5 | 62.1 | 58.1 | 55.4 | 43.0 | 17.7 | 89.6 | 87.6 |
| FashionIQ | 11.4 | 13.8 | 20.1 | 9.3 | 11.2 | 2.8 | 40.2 | 16.2 |
| Wiki-SS-NQ | 55.0 | 44.6 | 55.1 | 28.7 | 18.7 | 8.6 | 12.2 | 60.2 |
| OVEN | 41.1 | 45.0 | 56.0 | 39.5 | 1.6 | 5.9 | 69.4 | 56.5 |
| EDIS | 81.0 | 77.5 | 23.6 | 54.4 | 62.6 | 26.8 | 79.2 | 87.8 |
| *All Retrieval* | 53.0 | 52.3 | 31.6 | 33.9 | 35.4 | 11.5 | 61.8 | 67.4 |
| **Visual Grounding (4 tasks)** | | | | | | | | |
| MSCOCO | 33.8 | 34.5 | 46.4 | 28.9 | 22.1 | 10.8 | 46.6 | 80.6 |
| RefCOCO | 56.9 | 54.2 | 70.8 | 47.4 | 22.8 | 11.9 | 67.8 | 88.7 |
| RefCOCO-matching | 61.3 | 68.3 | 50.8 | 59.5 | 35.6 | 38.9 | 62.9 | 84.0 |
| Visual7W-pointing | 55.1 | 56.3 | 70.1 | 52.0 | 23.4 | 14.3 | 71.3 | 90.9 |
| *All Visual Grounding* | 51.8 | 53.3 | 59.5 | 47.0 | 26.0 | 19.0 | 65.3 | 86.1 |
| **Final Score (36 tasks)** | | | | | | | | |
| All | 37.8 | 39.7 | 34.8 | 25.2 | 27.8 | 13.3 | 44.7 | 62.9 |
| All IND | 37.1 | 39.3 | 32.3 | 25.3 | 31.0 | 14.9 | 47.1 | 67.5 |
| All OOD | 38.7 | 40.2 | 38.0 | 25.1 | 23.7 | 11.5 | 41.7 | 57.1 |

Table 7: Examples of datasets in MMEB (Part 1 of 4). *Instructions* are written in italic font style.

| Category | Dataset | Query Text | Query Image | Target Text | Target Image |
|---|---|---|---|---|---|
| | ImageNet-1K (Deng et al., 2009) | *Represent the given image for classification* | | Italian greyhound | - |
| | ImageNet-A (Hendrycks et al., 2021b) | *Represent the given image for classification.* | | sea anemone, anemone | - |
| | ImageNet-R (Hendrycks et al., 2021a) | *Represent the given image for classification.* | | baseball player | - |
| | N24News (Wang et al., 2021) | *Represent the given news image with the following caption for domain classification.* Ms. Goodman styled Amber Valletta with wings for a 1993 shoot by Peter Lindbergh for Harper's Bazaar. | | Style | - |
| Classification | VOC2007 (Everingham et al., 2014) | *Identify the object shown in the image.* | | bus | - |
| | SUN397 (Xiao et al., 2010) | *Identify the scene shown in the image.* | | firing range indoor | - |
| | ObjectNet (Barbu et al., 2019) | *Identify the object shown in the image.* | | mug | - |
| | Country-211 (Radford et al., 2021) | *Identify the country depicted in the image.* | | China | - |
| | HatefulMemes (Kiela et al., 2020) | *Represent the given image for binary classification to determine whether it constitutes hateful speech or not.* | | No | - |
| | Place365 (Zhou et al., 2017) | *Identify the scene shown in the image.* | | Airport Terminal | - |

Table 8: Examples of datasets in MMEB (Part 2 of 4). *Instructions* are written in italic font style.

| Category | Dataset | Query Text | Query Image | Target Text | Target Image |
|---|---|---|---|---|---|
| VQA | OK-VQA (Marino et al., 2019) | *Represent the given image with the following question.* What breed of dog is this? |  | chihuahua | - |
| | A-OKVQA (Schwenk et al., 2022) | *Represent the given image with the following question.* What is the metal basket near the net used to hold? |  | tennis balls | - |
| | DocVQA (Mathew et al., 2021) | *Represent the given image with the following question.* What is name of university? |  | university of california | - |
| | InfographicsVQA (Mathew et al., 2022) | *Represent the given image with the following question.* Which social platform has heavy female audience? |  | pinterest | - |
| | ChartQA (Masry et al., 2022) | *Represent the given image with the following question.* How many food item is shown in the bar graph? |  | 14 | - |
| | ScienceQA (Lu et al., 2022) | *Represent the given image with the following question.* Which of these states is farthest north? |  | South Carolina | - |
| | Visual7W-telling (Zhu et al., 2016) | *Represent the given image with the following question.* Where is the man sitting? |  | At the computer | - |
| | VizWiz (Gurari et al., 2018) | *Represent the given image with the following question.* Can you tell me what this medicine is please? |  | night time | - |
| | GQA (Hudson & Manning, 2019) | *Represent the given image with the following question.* What is under the utensil on the left? |  | The napkin is under the utensil. | - |
| | TextVQA (Singh et al., 2019) | *Represent the given image with the following question.* What is the brand of this camera? |  | dakota | - |

Table 9: Examples of datasets in MMEB (Part 3 of 4). *Instructions* are written in italic font style.

| Category | Dataset | Query Text | Query Image | Target Text | Target Image |
|---|---|---|---|---|---|
| Retrieval | VisDial (Das et al., 2017) | *Represent the given dialogue about an image, which is used for image retrieval.* Q:do you see a lot of people A:just 3 Q:what is the tennis player wearing A:white tennis dress Q:what color is her tennis racket A:black Q:is she wearing a hat A:a visor Q:is she close to the net A:no Q:do you see another player A:no Q:do you see a tennis bag A:no | - | *Represent the given image.* |  |
| | VisualNews-t2i (Liu et al., 2020) | *Retrieve an image of this news caption.* US goalkeeper Hope Solo makes a save. | - | *Represent the given image.* |  |
| | MSCOCO-t2i (Lin et al., 2014) | *Find me an everyday image that matches the given caption.* Man riding a motor bike on a dirt road on the countryside. | - | *Represent the given image.* |  |
| | WebQA (Chang et al., 2022) | *Find a Wikipedia image-passage pair that answers this question.* Do both the Hays County Courthouse in San Marcos, Texas and the Ike Wood House at 227 Mitchell Street in San Marcos, Texas have six columns on their front entrance? | - | *Represent the given Wikipedia image with related text information.* Hays County Courthouse (2018), San Marcos, TX The Hays County Courthouse in San Marcos, Texas. Listed on the National Register of Historic Places. 227 Mitchell, San Marcos, Texas Ike Wood House at 227 Mitchell Street in San Marcos, Texas. |  |
| | EDIS (Liu et al., 2023) | *Find a news image that matches the provided caption.* Tom Holland makes his debut in the Spidey suit in Captain America Civil War. | - | *Represent the given image with related text information.* Comic RiffsJon Favreau is set to reprise his Iron Man role for Spider Man: Homecoming. |  |
| | Wiki-SS-NQ (Ma et al., 2024a) | *Find the document screenshot that can answer the given query.* | - | *Represent the given document screenshot.* |  |
| | VisualNews-i2t (Liu et al., 2020) | *Find a caption for the news in the given photo.* |  | Canadian Prime Minister Stephen Harper shakes hands with President Obama during the North American Leaders Summit in Toluca Mexico in February 2014. | - |
| | MSCOCO-i2t (Lin et al., 2014) | *Find an image caption describing the given everyday image.* |  | A man on a bicycle riding next to a train. | - |

Table 10: Examples of datasets in MMEB (Part 4 of 4). *Instructions* are written in italic font style.

| Category | Dataset | Query Text | Query Image | Target Text | Target Image |
|---|---|---|---|---|---|
| Retrieval | CIRR (Liu et al., 2021) | *Given an image, find a similar everyday image with the described changes.* Show three bottles of soft drink. | | *Represent the given image.* | |
| | FashionIQ (Wu et al., 2021) | *Find an image to match the fashion image and style note.* Is shiny and silver with shorter sleeves and fit and flare. | | *Represent the given image.* | |
| | NIGHTS (Fu et al., 2023) | *Find a day-to-day image that looks similar to the provided image.* | | *Represent the given image.* | |
| | OVEN (Hu et al., 2023) | *Retrieve a Wikipedia image-description pair that provides evidence for the question of this image.* What is the name of this place? | | *Represent the given Wikipedia image with related text information.* Titisee. The Titisee is a lake in the southern Black Forest in Baden-Württemberg. It covers an area of 1.3 (km2) and is an average of 20 (m) deep. It owes its formation to the Feldberg glacier, the moraines of which were formed in the Pleistocene epoch and nowadays form the shores of the lake. The lake's outflow, at 840 (m) above sea level, is the River Gutach, which merges with the Haslach stream below Kappel to form the Wutach. The waters of the Titisee thus drain eventually into the Upper Rhine between Tiengen and Waldshut. On the north shore lies the. | |
| Grounding | MSCOCO (Lin et al., 2014) | *Select the portion of the image that isolates the object of the given label* The lable of the object is "stop sign". | | *Represent the given cropped image of the object.* | |
| | Visual7W-Pointing (Zhu et al., 2016) | *Select the portion of the image that answers the given question.* Which door is behind a person sitting on a bench? | | *Represent the given cropped image of the object.* | |
| | RefCOCO (Kazemzadeh et al., 2014) | *Select the portion of the image that follows the language expressions.* man in black coat | | *Represent the given cropped image of the object.* | |
| | RefCOCO-Matching (Kazemzadeh et al., 2014) | *Select the portion of the image that follows the language expressions.* kid on right in back, blondish hair | | *Select the portion of the image that follows the language expressions.* top right kid | |

Table 11: Zero-shot text-image retrieval performance on Flickr30K. As a general multimodal representation model, VLM2VEC can still achieve competitive T2I (Text-to-Image) and I2T (Image-to-Text) scores when compared to existing CLIP-like models. The baseline numbers are sourced from Sun et al. (2023) and Zhang et al. (2024). VLM2VEC is built upon the LLaVA-1.6 backbone.

| Model | image retrieval | | | text retrieval | | |
|---|---|---|---|---|---|---|
| | R@1 | R@5 | R@10 | R@1 | R@5 | R@10 |
| OpenAI CLIP-B/16 | 62.1 | 85.6 | 91.8 | 81.9 | 96.2 | 98.8 |
| Open CLIP-B/16 | 69.8 | 90.4 | 94.6 | 86.3 | 97.9 | 99.4 |
| EVA-02-CLIP-B/16 | 71.2 | 91.0 | 94.7 | 85.7 | 96.7 | 98.9 |
| OpenAI CLIP-L/14 | 65.2 | 87.3 | 92.0 | 85.2 | 97.3 | 99.0 |
| Open CLIP-L/14 | 75.0 | 92.5 | 95.6 | 88.7 | 98.4 | 99.2 |
| EVA-02-CLIP-L/14 | 77.3 | 93.6 | 96.8 | 89.7 | 98.6 | 99.2 |
| MagicLens-B | 76.2 | 93.7 | 96.5 | 87.9 | 97.7 | 99.5 |
| MagicLens-L | 79.7 | 95.0 | 97.4 | 89.6 | 98.7 | 99.4 |
| VLM2VEC (LLaVA-1.6) | **80.3** | **95.0** | **97.4** | **94.6** | **99.5** | **99.8** |

