# OpenReview forum: "VLM2Vec: Training Vision-Language Models for Massive Multimodal Embedding Tasks"
_ICLR.cc/2025/Conference — ICLR 2025 Poster_

### Official Review · Reviewer_z3VP · 2024-11-03

**Soundness:** 3
**Presentation:** 4
**Contribution:** 3
**Rating:** 8
**Confidence:** 5

**Summary:**

This paper presents a multimodal embedding framework and introduces a comprehensive dataset, MMEB, which includes 36 datasets across four meta-task categories, specifically designed for training and evaluating embedding models. Additionally, the paper proposes VLM2VEC, a contrastive training framework that adapts an existing vision-language model into an embedding model. Experimental results indicate that the proposed model achieves 10%~20% improvement over previous multimodal embedding models, on both in-distribution and out-of-distribution datasets.

**Strengths:**

1. The method introduced in this paper is robust. With the authors' commitment to open- sourcing both the model and data, the proposed approach will to be valuable for a wide range of applications.
2. The experiments are thorough, effectively demonstrating the model's efficacy.
3. The paper is well-structured and clearly written.

**Weaknesses:**

1. The method proposed in this paper is relatively straightforward. Its performance, however, varies significantly across different tasks. Would suggest including more comparative experiments to explore and clarify the reasons behind these performance differences across tasks.

**Questions:**

n/a

---

> ### Author Response · Authors · 2024-11-25
> **Author Response**
>
> Thank you so much for your constructive feedback to our work! The following is our response:
>
> **W1: Relatively straightforward Approach**
>
> Thank you for your feedback and for acknowledging the straightforwardness of our proposed method. We agree that our framework is conceptually simple, and this simplicity is an advantage here. Our framework is highly scalable and generalizable to a wide range of VLMs. Our experiments also showcase the ability of VLM2Vec to adapt across diverse meta-tasks and domains, underlining its robustness as a multimodal embedding framework.
>
> **W2: Clarify the reasons behind these performance differences**
>
> We tested the embedding model on a variety of 36 tasks including classification, VQA, retrieval and grounding. These tasks are sourced from various existing datasets. Due to the task type, their difficulty levels vary significantly.
> For example, ImageNet-A is an adversarially constructed set, which poses great challenges to all the models. In contrast, VOC2007 is a traditional classification task used to test basic visual perception, where many models can reach very high accuracy. Similarly, retrieval tasks like FashionIQ and CIRR  require understanding nuances between different fashion images, which pose great challenges to all the models and lead to relatively low performance.
>
> As part of our future work, we plan to conduct detailed analyses of the model’s performance and behaviors across different domains and complexities. This will include exploring its handling of abstract versus concrete concepts, sensitivity to the granularity of cropping, and its ability to represent multiple objects or process multiple images. These directions will help us gain a deeper understanding of the task-specific strengths and limitations of the proposed framework.
>
> Thank you again for your constructive suggestions, and we will aim to incorporate more detailed task-specific evaluations in future iterations.

---

### Official Review · Reviewer_hG4f · 2024-11-03

**Soundness:** 2
**Presentation:** 2
**Contribution:** 2
**Rating:** 3
**Confidence:** 5

**Summary:**

This paper focuses on the development of universal multimodal embedding models. The authors introduce the Massive Multimodal Embedding Benchmark (MMEB), which comprises classification, visual question answering (VQA), retrieval, and grounding tasks. They propose a contrastive training framework, VLM2Vec, to transform VLM into an embedding model. Experimental results demonstrate that VLM2Vec achieves promising performance on the MMEB.

**Strengths:**

1. The writing is clear and easy to understand. This paper is easy to follow.
2. The constructed benchmark, MMEB, covers a wide range of tasks and is valuable.

**Weaknesses:**

1. The experiments presented are insufficient, as the authors limit their evaluation to the proposed MMEB benchmarks using only the precision metric. To provide a more comprehensive comparison with other clip-like models, it is crucial to assess the embedding model on some traditional benchmarks, such as ELEVATER and zero-shot retrieval on COCO and Flickr30k.
2. The evaluation of out-of-distribution may be unfair. Current MLLMs are usually fine-tuned on various downstream datasets. Most of datasets that construct MMEB are usually used in the fine-tuning stage of MLLMs. Since Phi-3.5-V doesn't provide details of fine-tuning datasets,  it is hard to evaluate the out-of-distribution capabilities of VLM2Vec.
3. Lack of some experimental details. The authors need to provide details of the training of the comparative models on the MMEB in-domain datasets. Additionally, it is unfair to compare CLIP without instructions. Although the authors claim that based on other related work, introducting instructions to CLIP would not perform well, Table 4 shows that instructions have a significant impact, and the CLIP model should also have experimental results under the same settings.

**Questions:**

1. What do the x-axis and legend in Figure 5 represent? What do the three subplots represent respectively?
2. Does this training method affect the original capabilities of MLLM? I'm curious about the performance of VLM2Vec on MLLM-related benchmarks, such as MMBench and MME.

---

> ### Author Response · Authors · 2024-11-24
> **Response to Reviewer hG4f (Part 1)**
>
> Thank you for your valuable comments and feedback.
>
> We would like to emphasize that our primary goal was to introduce a unified benchmark for evaluating text-image embedding models, providing a more comprehensive perspective on how current embedding models perform across various tasks and domains. Additionally, we aimed to explore the potential of adapting pretrained VLMs to embedding tasks. As demonstrated in Section 4.4, our scaling experiments reveal that further scaling significantly improves model performance, and we are still far from reaching the limits of this scaling trend. While we are excited to highlight that our current model outperforms previous approaches on MMEB, we also aim to share intriguing findings on effectively tuning VLMs into embedding models. These include insights on LoRA versus full fine-tuning, the impact of scaling, and the interactions between meta-tasks.
>
> We address the raised concerns as follow:
>
> **W1.**
> >On Evaluating Traditional Benchmarks.
>
> We acknowledge the importance of comparing performance on well-established zero-shot benchmarks for a broader perspective. Since we included MSCOCO in our training split, we focus here on presenting zero-shot retrieval results on Flickr30K, as shown in the table below. Our VLM2Vec demonstrates strong performance, achieving competitive T2I (Text-to-Image) and I2T (Image-to-Text) scores compared to existing CLIP-like models. It is worth noting that VLM2Vec is specifically designed for instructional text/image representation rather than being optimized solely for retrieval tasks.
>
> We believe that by incorporating additional retrieval-specific training data and techniques such as hard negative mining, the retrieval performance of VLM2Vec can be further enhanced. Thank you for highlighting this, and we will expand on this aspect in the revised version to strengthen comparisons across traditional benchmarks. We also plan to include additional ranking-based metrics, such as MRR and NDCG, when presenting the MMEB leaderboard in the future.
>
> | Model             | Recall@1 (T -> I) | Recall@5 (T -> I) | Recall@10 (T -> I) | Recall@1 (I -> T) | Recall@5 (I -> T) | Recall@10 (I -> T) |
> |-------------------|--------------|--------------|---------------|--------------|--------------|---------------|
> | OpenAI CLIP-B/16  | 62.1         | 85.6         | 91.8          | 81.9         | 96.2         | 98.8          |
> | Open CLIP-B/16    | 69.8         | 90.4         | 94.6          | 86.3         | 97.9         | 99.4          |
> | EVA-02-CLIP-B/16  | 71.2         | 91.0         | 94.7          | 85.7         | 96.7         | 98.9          |
> | OpenAI CLIP-L/14  | 65.2         | 87.3         | 92.0          | 85.2         | 97.3         | 99.0          |
> | Open CLIP-L/14    | 75.0         | 92.5         | 95.6          | 88.7         | 98.4         | 99.2          |
> | VLM2Vec           | 74.1         | 92.8         | 95.9          | 88.8         | 98.7         | 99.5          |

---

> ### Author Response · Authors · 2024-11-24
> **Response to Reviewer hG4f (Part 2)**
>
> **W2**
> >On Out-of-Distribution (OOD) Evaluation.
>
> Thank you for raising the concern regarding potential OOD definition during the evaluation of Phi-3.5-vision. To address this issue, we additionally trained VLM2Vec using the llava-hf/llava-v1.6-mistral-7b-hf backbone, which provides a transparent training recipe. According to its documentation, among our OOD evaluation sets,  only RefCOCO was used during Llava training. So the other OOD sets are still valid. We have included the corresponding results in the table below (also will be added to the revised paper).
>
> Both VLM2Vec models were trained under the same settings (LoRA-8, 2k steps, batch size of 1k), with one key difference in image processing. Due to time constraints, all input images for llava-v1.6-mistral training were resized to 336x336 to avoid inconsistencies caused by dynamic cropping and to accelerate training. While this resizing significantly limits the model's visual understanding capabilities, leading to performance drops across tasks, VLM2Vec trained with llava-v1.6-mistral still outperforms baseline models on most tasks.
>
> Overall, VLM2Vec (Llava-1.6) consistently underperforms VLM2Vec (Phi-3.5) on IND and OOD.  For example, llava-1.6 has been trained on the training set of our several IND datasets like OKVQA, DocVQA, ChartVQA, its VLM2Vec performance is still lagging behind Phi-3.5 on these sets. This indicate that the gap is unlikely due to the instruction tuning set overlap, but rather from the models' underlying perception and grounding capabilities.
>
> Our goal is to provide the VLM2Vec framework and adapt it to all SOTA VLMs for representation learning in the future.
>
>
> | Model                      | Classification | VQA  | Retrieval | Grounding | IND  | OOD  | Overall |
> |----------------------------------------|----------------|------|-----------|-----------|------|------|---------|
> | CLIP                                       | 42.8          | 9.1  | 53.0      | 51.8      | 37.1 | 38.7 | 37.8    |
> | SigLIP                                    | 40.3          | 8.4  | 31.6      | 59.5      | 32.3 | 38.0 | 34.8    |
> | OpenCLIP                              | 47.8          | 10.9 | 52.3      | 53.3      | 39.3 | 40.2 | 39.7    |
> | UniIR                                      | 42.1          | 15.0 | 60.1      | 62.2      | 44.7 | 40.4 | 42.8    |
> | VLM2Vec(Phi-3.5v)                | 54.8          | 54.9 | 62.3      | 79.5      | 66.5 | 52.0 | 60.1    |
> | VLM2Vec(llava-v1.6-mistral)  | 54.7          | 50.3 | 56.2      | 64.0      | 61.0 | 47.5 | 55.0    |
>
> **W3.1**
> >On Experimental Details for Baselines.
>
> The baseline models are trained with different data setups. For example, CLIP is trained on image-text pairs, E5-v is trained using text data, and UniIR is trained on retrieval tasks. One of the key contributions of our paper is training on diverse setups:
> 1. Diverse combinations of image and text modalities: Both the query and target can be an image, text, or a combination of both.
> 2. Diverse meta-tasks: We incorporate four meta-tasks and use 20 datasets, which sets our approach apart from models like UniIR that are trained on a single meta-task (retrieval).
>
> Furthermore, to highlight the advantages of using VLM as a backbone for fusing text and image data in representation learning, we retrained CLIP-based models on our in-domain dataset.
>
> For the remaining baseline models, UniIR and MagicLens also utilize a shallow fusion approach based on CLIP models, with their primary contribution being the datasets they were trained on. Therefore, we have not included the fine-tuned versions of these two models in this comparison. E5-v is a contemporary work that also leverages VLM as its backbone but is trained exclusively on text data. Therefore, it is not appropriate to fine-tune their model on our datasets.
>
> | Model          | Classification  | VQA  | Retrieval  | Grounding   | IND  | OOD  | AVG  |
> |----------------|------|------|------|------|------|------|------|
> | CLIP (No FineTune)   | 42.8 | 9.1  | 53.0 | 51.8 | 37.1 | 38.7 | 37.8 |
> | OpenCLIP (No FineTune)| 47.8 | 10.9 | 52.3 | 53.3 | 39.3 | 40.2 | 39.7 |
> | CLIP (FineTune)      | 55.2 | 19.7 | 53.2 | 62.2 | 47.6 | 42.8 | 45.4 |
> | OpenCLIP ( FineTune)  | 56.0 | 21.9 | 55.4 | 64.1 | 50.5 | 43.1 | 47.2 |
> | VLM2Vec        | 54.8 | 54.9 | 62.3 | 79.5 | 66.5 | 52.0 | 60.1 |

---

> ### Author Response · Authors · 2024-11-24
> **Response to Reviewer hG4f (Part 3)**
>
> **W3.2**
> >CLIP w/ instructions
>
> Thanks for your suggestion! We have provided the results of the CLIP model using instructions here. Since the CLIP-family model is not well-suited for instruction-following and does not fuse text and image information as effectively as VLM2Vec, using instructions can lead to a decline in performance.
> | Model                   | Classification  | VQA  | Retrieval  | Grounding   | IND  | OOD  | AVG  |
> |-------------------------|------|------|------|------|------|------|------|
> | CLIP (w/o Instruction)  | 42.8 | 9.1  | 53.0 | 51.8 | 37.1 | 38.7 | 37.8 |
> | CLIP (w/ Instruction)   | 17.4 | 8.0  | 41.3 | 52.9 | 23.8 | 30.3 | 26.7 |
>
> **Q1**
> >Figure 5 details
>
> Figure 5 illustrates the generalizability of the three models on unseen meta-tasks. Taking the first subplot as an example, the legend indicates that the model is trained on only one meta-task. For instance, "VLM2Vec_VQA" means the model is trained exclusively on the VQA meta-task, while "VLM2Vec_RET" indicates training on the retrieval meta-task. The x-axis represents the meta-task being evaluated. For example, "VLM2Vec_VQA" achieves a precision of 35.9 on the classification task. Figure 5 demonstrates that the retrieval task provides the best generalization ability to our model. This result is intuitive since the retrieval task dataset includes more diverse instructions and combinations of  text and visual modalities.
>
> The table below provides a detailed numerical breakdown of the same data shown in Figure 5, offering an alternative perspective for detailed comparison and analysis.
>
> | Train Task | All  | Classification  | VQA  | Retrieval  | Grounding    |
> |------------|------|------|------|------|------|
> | TrainAll   | 52.0 | 50.4 | 46.4 | 52.6 | 68.6 |
> | TrainCLS   | 26.9 | 53.6 | 12.6 | 13.8 | 35.1 |
> | TrainVQA   | 32.6 | 35.9 | 52.8 | 13.0 | 33.0 |
> | TrainRET   | 37.4 | 29.1 | 14.2 | 59.1 | 51.3 |
>
> **Q2**
> >Whether this training method affect the original capabilities of MLLM
>
> This is indeed a very valid question. In our VLM2Vec-Lora version, we only need to load 30MB low-rank adapter weights to activate Phi-3.5's retrieval capabilities. In the generative scenario, we simply unload the low-rank adapter to recover the original checkpoint to maintain its performance. This essentially makes the model very flexible to function as both a retriever and generator. In the multimodal RAG setup, we can load both the phi-3.5 checkpoint and lora in the memory. In the retrieval stage, we will load the LoRA module to retrieve multimodal information. After that, we can unload the LoRA module to read the retrieved information and maintain its generative performance.
>
> It's also worth highlighting that recent research has proposed training strategies enabling a single model to handle both generative and embedding tasks in text-based language problems (e.g., https://arxiv.org/pdf/2402.09906). Exploring such approaches in the multimodal setting is a promising direction, and we see this as a natural extension of our work. We will aim to present results on the benchmarks you mentioned in the near future. Thank you again for the valuable feedback!

---

> ### Author Response · Authors · 2024-11-26
> **Further clarification of W3.1 (Details of the training of the comparative models on the MMEB in-domain datasets.)**
>
> We realize that we may have misunderstood the first part of W3, and our previous response might have caused some confusion, so we would like to provide further clarifications here. We used four categories of models (CLIP-family, UniIR, MagicLens, and E5-V) as baselines. In the initial version of our paper, we evaluated these models on the MMEB evaluation datasets without any fine-tuning. By comparing their performance with our VLM2Vec on the 16 OOD datasets, we demonstrated that VLM2Vec outperforms these baselines.
> The superior performance of VLM2Vec stems from two key contributions of our work:
> 1. The training data and training strategy.
> 2. The framework uses VLM as the backbone of the embedding model, enabling a deep fusion of image and text features, as opposed to the shallow fusion used by other CLIP-based models.
>
> As suggested by one of the reviewers, it would be fairer to fine-tune the baseline models on our MMEB training data. We agree with this valid point. Given that the second contribution of VLM2Vec (deep fusion of text & image features through VLM) is pivotal to its strong performance, this comparison (VLM2Vec vs. fine-tuned CLIP-based embedding models) could better highlight the advantage of our framework. In our revised version, we fine-tuned some of the baseline models on the same MMEB training datasets, using exactly the same training setup (e.g., batch size, number of training steps). Below are the reasons why we chose certain baseline models for fine-tuning and why we excluded others.
> * For the CLIP-family category, we fine-tuned the two models that achieved the best performance on MMEB eval prior to fine-tuning (CLIP and OpenCLIP).
> * UniIR and MagicLens are both CLIP-based models that incorporate a shallow fusion layer to combine text and image information. The primary contributions of these models lie in their training data: UniIR uses diverse types of retrieval data, while MagicLens is trained on millions of web image pairs. Since their architecture is similar to the first category and their main contribution is actually their own training data, we decided not to fine-tune them.
>
> In conclusion, VLM2Vec's strong performance can be attributed to two key factors:
> 1. The training data and training strategy.
> 2. The deep fusion of image and text features by leveraging VLMs as backbones.
>
> **In the initial version of our work, we did not fine-tune other embedding models on our own data.
> In our revised version, we added additional fine-tuning experiments to highlight the significance of the second point.**
> Please let us know whether we have adequately addressed your concern.

---

> > ### Comment · Reviewer_hG4f · 2024-12-01
> >
> > Thank you for your feedback.
> >
> > **Weakness 1:**
> >
> > 1. Are these results obtained from phi-3.5v ?
> > 2. Generally, cross-modal retrieval tasks are related to the style of the training text. The text style of flickr30k is simple descriptive sentences, which is completely different from the style of the instructional data used for training. Why can VLM2Vec still achieve exceptional performance? This is a rather counterintuitive phenomenon. Could the authors explain the reason behind this phenomenon?
> >
> > 3. Could the authors provide the results of zero-shot classification on ELEVATER? This is a very important indicator for embedding models.
> >
> >
> >
> > **Weakness 2:**
> >
> > The comparison is conducted under an unfair setting (with models of different sizes, training data, etc.), and the performance change from phi-3.5-v to llava-v1.6 does not provide enlightening insights. A more appropriate comparison would be against training llava-v1.6 with out-of-distribution (OOD) data, in order to reflect the impact of data leakage on evaluation.
> >
> > The primary concern here is that MMEB utilizes open-source data, a significant portion of which may have been involved in the training process of MLLMs. For retrieval tasks, data leakage can lead to unfair comparisons, thereby restricting the usability of MMEB and potentially preventing the evaluation of many MLLMs.

---

> ### Author Response · Authors · 2024-11-27
> **Rebuttal followup**
>
> Dear Reviewer hG4f,
>
> We would like to learn if our response addresses your concerns and questions, and we invite any additional feedback or thoughts for improving our paper. If you feel that our responses resolve the issues raised, we would be grateful if you could consider reflecting this in the evaluation. We would be happy to address any further concerns or questions. Thank you again for your time and effort!

---

> > ### Author Response · Authors · 2024-11-30
> > **Rebuttal followup**
> >
> > Dear Reviewer hG4f,
> >
> > Happy Thanksgiving! We really appreciate your feedback on our paper. We would like to learn if our response addresses your concerns and questions, and we invite any additional feedback or thoughts for improving our paper.

---

> > > ### Author Response · Authors · 2024-11-30
> > > **Rebuttal followup**
> > >
> > > Dear Reviewer hG4f,
> > >
> > > As the discussion phase draws to the end, we kindly invite you to follow up on the discussion, and we would be happy to address any additional concerns or questions you might have.

---

> ### Author Response · Authors · 2024-12-01
> **Response**
>
> > 1. Are these results obtained from phi-3.5v ?
>
> Yes, the Flickr30K results are obtained from Phi-3.5v.
>
> > 2.  Why can VLM2Vec still achieve exceptional performance?
>
> We do not think this result is counterintuitive. The strong performance is achieved through generalization from the instructional training setup, consistent with findings from prior works such as Flan [1] and UniIR [2]. We incorporate instructions into Flickr30K queries (the instructions we used are "Retrieve an image that matches the given caption:" and "<|image_1|>" Find an image caption describing the given image"). Additionally, our training dataset includes similar caption-image pairs, such as those in VisualNews (T2I and I2T). The model learns to generalize across tasks by following task-specific instructions, demonstrating its ability to adapt effectively to different retrieval tasks.
>
> [1] Chung, Hyung Won, et al. "[Scaling instruction-finetuned language models](https://arxiv.org/abs/2210.11416)." JMLR 25.70 (2024): 1-53.
>
> [2] Wei, Cong, et al. "[Uniir: Training and benchmarking universal multimodal information retrievers](https://arxiv.org/pdf/2311.17136)." arXiv preprint arXiv:2311.17136 (2023).
>
> > 3. Could the authors provide the results of zero-shot classification on ELEVATER?
>
> ELEVATER contains several datasets that overlap with those already included in MMEB-eval (e.g., HatefulMemes, Country-211, VOC2007). Given its size, we will aim to present the results on ELEVATER in the future updates as best as we can.
>
> > 4.  A more appropriate comparison would be against training llava-v1.6 with out-of-distribution (OOD) data ...
>
> Could you please elaborate on this suggestion? To clarify, **all VLM2Vec models (including Llava-v1.6) are not trained on any OOD datasets**. Sixteen OOD datasets were intentionally excluded from training and reserved for evaluation purposes. Regarding Llava-v1.6 specifically, we have verified its training recipe: it only used RefCOCO as an OOD dataset during its training, which is also included in the MMEB OOD split. Therefore, the VLM2vec-Llava-v1.6 results for the other 15 OOD datasets remain valid, and the model demonstrates strong generalization performance on unseen data.
>
> > 5. For retrieval tasks, data leakage can lead to unfair comparisons.
>
> Could you clarify where the alleged data leakage originates? It is important to note that data leakage specifically refers to scenarios where the test set or its labels are exposed during training. Similarities between the training set and OOD data in terms of task or domain characteristics do not constitute leakage unless identical data points are present. Without clear evidence, this allegation does not hold.
>
> If you are instead referring to distributional overlap (e.g., the training data shares similar tasks/domains with OOD data but does not contain identical samples), this concern can be addressed. The updated VLM2Vec-Llava-v1.6 results show that the model, which only used RefCOCO during training, performs strongly across the remaining 15 OOD datasets. This demonstrates robust generalization capabilities despite not being trained on those datasets.
>
> Thank you for your feedback, and we look forward to further clarification on these points.

---

> > ### Author Response · Authors · 2024-12-01
> > **Followup**
> >
> > Dear Reviewer hG4f,
> >
> > Thanks for your feedback. Would you mind reading our latest response?

---

> ### Author Response · Authors · 2024-12-02
> **Additional Results for VLM2Vec with the LLava-1.6 Backbone**
>
> We have trained an improved version of the VLM2Vec-llava-1.6. The primary difference from the previous version is the adoption of higher-resolution images, increasing from 336x336 to 1344x1344. We would like to share the results here. **We believe that VLM2Vec-Llava-1.6, which leverages LLava-1.6 with a transparent pre-training data recipe, demonstrates the effectiveness of our VLM2Vec framework and its strong generalization ability on OOD evaluation datasets.**
>
> >Flickr-30K result
>
> VLM2Vec-Llava-1.6 has demonstrated strong performance on image-text retrieval tasks, despite not being specifically trained for retrieval.
>
> | Model             | R@1 (T -> I) | R@5 (T -> I) | R@10 (T -> I) | R@1 (I -> T) | R@5 (I -> T) | R@10 (I -> T) |
> |-------------------|--------------|--------------|---------------|--------------|--------------|---------------|
> | OpenAI CLIP-B/16  | 62.1         | 85.6         | 91.8          | 81.9         | 96.2         | 98.8          |
> | Open CLIP-B/16    | 69.8         | 90.4         | 94.6          | 86.3         | 97.9         | 99.4          |
> | EVA-02-CLIP-B/16  | 71.2         | 91.0         | 94.7          | 85.7         | 96.7         | 98.9          |
> | OpenAI CLIP-L/14  | 65.2         | 87.3         | 92.0          | 85.2         | 97.3         | 99.0          |
> | Open CLIP-L/14    | 75.0         | 92.5         | 95.6          | 88.7         | 98.4         | 99.2          |
> | VLM2Vec  (phi-3.5-v)       | 74.1         | 92.8         | 95.9          | 88.8         | 98.7         | 99.5         |
> | VLM2Vec  (llava-1.6)       | 80.3         | 95.0         | 97.4          | 94.6         | 99.5         | 99.8         |
>
>
> >MMEB result
>
> FFT means fully fine-tuned on the MMEB train.
> The results clearly show that VLM2Vec-Llava-1.6 exhibits strong generalization ability on OOD datasets, further proving the effectiveness of our VLM2Vec framework.
>
> | Model                | Classification | VQA   | Retrieval | Grounding | IND   | OOD   | Overall |
> |----------------------|----------------|-------|-----------|-----------|-------|-------|---------|
> | CLIP                | 42.8          | 9.1   | 53.0      | 51.8      | 37.1  | 38.7  | 37.8    |
> | CLIP (FFT)          | 55.2          | 19.7  | 53.2      | 62.2      | 47.6  | 42.8  | 45.4    |
> | OpenCLIP            | 47.8          | 10.9  | 52.3      | 53.3      | 39.3  | 40.2  | 39.7    |
> | OpenCLIP (FFT)      | 56.0          | 21.9  | 55.4      | 64.1      | 50.5  | 43.1  | 47.2    |
> | VLM2Vec (Phi-3.5v)  | 54.8          | 54.9  | 62.3      | 79.5      | 66.5  | 52.0  | 60.1    |
> | VLM2Vec (llava-v1.6)| 61.2          | 49.9  | 67.4      | 86.1      | 67.5  | 57.1  | 62.9    |

---

> > ### Author Response · Authors · 2024-12-02
> > **Clarification on why we use LLava-1.6**
> >
> > It seems there might be a misunderstanding regarding the second weakness you mentioned.
> >
> > First, we want to clarify that our goal is not to compare Phi-3.5-v and LLava-1.6. Instead, our goal is to develop a framework that leverages any VLM as a general representation model, which is the core idea behind VLM2Vec. We fine-tuned VLMs only on MMEB in-domain datasets and evaluated them on MMEB out-of-domain datasets.
> >
> > You previously suggested that some of our MMEB OOD datasets might already be included in the training process of certain VLMs. Phi-3.5-v does not provide a clear description of the data it was trained on, which is why we chose LLava-1.6. LLava-1.6 has a transparent pre-training data recipe and nearly no overlap with our MMEB OOD datasets. The strong results achieved by VLM2Vec-LLava-1.6 demonstrate that these outcomes are not because LLava-1.6 had prior exposure to the OOD datasets. Instead, they highlight the effectiveness of our VLM2Vec framework in providing a good general multimodal representation.
> >
> > We hope this explanation addresses your concerns.

---

> > > ### Author Response · Authors · 2024-12-02
> > > **Final Day Reminder and Hope for Your Reply**
> > >
> > > Dear Reviewer hG4f,
> > >
> > > This is a kind reminder that today is the final day of the discussion period. We greatly appreciate your time, efforts, and suggestions for our paper. Please let us know if our responses have addressed your concerns and questions.

---

### Official Review · Reviewer_kzgZ · 2024-11-04

**Soundness:** 2
**Presentation:** 3
**Contribution:** 3
**Rating:** 5
**Confidence:** 4

**Summary:**

This paper proposes a novel framework called VLM2Vec for training vision-language models into embedding models capable of handling massive multimodal embedding tasks. This paper proposes MMEB, a benchmark with 36 datasets spanning classification, visual question answering, retrieval, and visual grounding. VLM2Vec deeply integrates visual and textual features, enabling it to capture cross-modal relationships more effectively.

**Strengths:**

1. Extensive experiments are conducted using multiple datasets and baselines, demonstrating the effectiveness of the proposed framework.
2. The development of the MMEB is original, as it provides a comprehensive framework for training and evaluating embedding models across diverse tasks and modalities.
3. Detailed ablation studies are performed to analyze the impact of various hyperparameters and training strategies on the model's performance.

**Weaknesses:**

Here are my constructive insights on how the paper could improve:
1. Benchmark Design: The number of candidates for each retrieval task in the MMEB benchmark is limited to 1000, which may not accurately reflect real-world scenarios where the number of candidates could be much larger. Increasing the number of candidates would make the benchmark more realistic and challenging.
2. Instead of randomly selecting 1000 candidates for each task, the standard test sets provided by the original datasets should be used. This would ensure a fairer comparison with other methods that have been evaluated on these datasets.
3. Fair Comparison: Table 2 compares VLM2Vec with baseline methods that have not been trained on the MMEB training sets. For a more accurate comparison, it would be insightful to retrain the baseline methods on the same training sets and then compare their performance with VLM2Vec.
4. Baseline Evaluation: The paper mentions UniIR but does not include results for the UniIR-CLIP version, which according to the UniIR paper, performs better than the UniIR-BLIP version. Including these results would provide a more comprehensive comparison.
5. The MMEB benchmark reformulates classification, VQA, retrieval, and visual grounding tasks as ranking problems. While ranking-based formulations can be useful for classification and retrieval tasks, applying the same approach to VQA and visual grounding might not be ideal. These tasks often require more nuanced understanding and reasoning. Considering alternative formulations, such as direct answer generation for VQA and object localization for visual grounding, might yield more meaningful results and better reflect the nature of these tasks.

**Questions:**

Refer to weaknesses.

---

> ### Author Response · Authors · 2024-11-24
> **Response to Reviewer kzgZ (Part 1)**
>
> We deeply appreciate Reviewer kzgZ’s insightful comments, which guided us to enrich our benchmark setting and add more analysis to show the strength of VLM2Vec model. We respond to Reviewer kzgZ’s comments as follows with more analytical results.
>
> **W1**
> >Benchmark design with an increased number of candidates.
>
> We agree that increasing the number of candidates would make the benchmark more realistic and challenging. But we also considered computation cost when designing the benchmark. We were concerned that having as many as a million candidates would lead to very low adoption of the benchmark due to the massive computation cost.
> Choosing 1000 as the number of candidates is a tradeoff between practicality and computation cost. For our VLM2Vec model, evaluation takes approximately 5 hours on 4 H100 GPUs, which is reasonable for most researchers. Additionally, we plan to release a more challenging version of MMEB with a larger number of candidates and increased difficulty. We investigate the performance of models with different numbers of candidates in a set of (100, 500, 1000, 2000, 5000), and choose 1000 as it strikes a balance between unsaturated benchmark performance and reduced iteration time for users.
>
>
> | #Candidates | Classification | VQA  | Retrieval | Grounding  | IND   | OOD   | Overall   |
> |------------|---------|---------|-------|--------|---------|--------|---------|
> | 100        | 54.8  | 81.8  | 86.1  | 89.6  | 85.2  | 65.9  | 76.6  |
> | 500        | 54.8  | 65.9  | 72.6  | 82.8  | 74.6  | 57.3  | 66.9  |
> | 1000       | 54.8  | 54.9  | 62.3  | 79.5  | 66.5  | 52.0  | 60.1  |
> | 2000       | 54.8  | 50.1  | 56.7  | 71.0  | 62.2  | 48.0  | 55.9  |
> | 5000       | 54.8  | 41.3  | 46.5  | 65.3  | 54.5  | 43.2  | 49.5  |
>
> (In classification, the number of candidates is fixed, which is the number of classes pre-defined.)
>
> **W2**
> >Including original test splits for fair comparison
>
> We agree that retaining the original test splits can facilitate comparisons with existing benchmark scores, and this was indeed considered during the initial draft of our benchmark. One issue we encountered is that some original benchmarks have large amounts of test examples and candidates, which can incur massive computation costs. This is a lesson we learned from existing text embedding benchmarks: MTEB and BEIR’s original test set comes at the expense of lengthy evaluation times—MTEB, for example, can take several weeks of GPU hours to complete. To address this, we adopted downsampling as a tradeoff, allowing us to significantly reduce inference costs while maintaining reasonable quality. This approach is also used in contemporary benchmarks like MMTEB.
> Since our primary goal is to create a comprehensive testbed that evaluates models across a wide range of meta-tasks and domains. To achieve this, we prioritized task and domain coverage by adopting a unified setup for all MMEB tasks, including a ranking formulation and resampled query/target pairs, to streamline benchmarking efforts for users.
>
> **W3**
> >Baseline methods trained on MMEB training sets
>
> We retrain several baseline models from the CLIP family using our in-domain data  and present the results here to demonstrate that VLM achieves better fusion of text and image information and potentially offers improved generalization capabilities.
> For the remaining baseline models, UniIR and MagicLens also utilize a shallow fusion approach based on CLIP models, with their primary contribution being the datasets they were trained on. Therefore, we have not included the fine-tuned versions of these two models in this comparison. E5-v is a contemporary work that also leverages VLM as its backbone but is trained exclusively on text data. Therefore, it is not appropriate to fine-tune their model on our datasets.
>
> | Model          | Classification | VQA  | Retrieval | Grounding   | IND  | OOD  | AVG  |
> |----------------|------|------|------|------|------|------|------|
> | CLIP (No FineTune)   | 42.8 | 9.1  | 53.0 | 51.8 | 37.1 | 38.7 | 37.8 |
> | OpenCLIP (No FineTune)| 47.8 | 10.9 | 52.3 | 53.3 | 39.3 | 40.2 | 39.7 |
> | CLIP (FineTune)      | 55.2 | 19.7 | 53.2 | 62.2 | 47.6 | 42.8 | 45.4 |
> | OpenCLIP ( FineTune)  | 56.0 | 21.9 | 55.4 | 64.1 | 50.5 | 43.1 | 47.2 |
> | VLM2Vec        | 54.8 | 54.9 | 62.3 | 79.5 | 66.5 | 52.0 | 60.1 |

---

> ### Author Response · Authors · 2024-11-24
> **Response to Reviewer kzgZ (Part 2)**
>
> **W4.**
> >Baseline Evaluation Using an Improved Version of the UniIR Model.
>
> Thank you for the suggestion! We evaluated the best version of the UniIR model (CLIP-SF) and have included the results in the table below. While CLIP-SF performed slightly better than the BLIP-FF version, it still falls significantly behind VLM2Vec across all metrics. We will include these results in our revised paper to ensure a more comprehensive comparison.
>
>
> | Model   | Classification | VQA  | Retrieval | Grounding  | IND  | OOD  | AVG  |
> |---------|------|------|------|------|------|------|------|
> | BLIP-FF | 42.1 | 15.0 | 60.1 | 62.2 | 44.7 | 40.4 | 42.8 |
> | CLIP-SF | 44.3 | 16.2 | 61.8 | 65.3 | 47.1 | 41.7 | 44.7 |
> | VLM2Vec | 54.8 | 54.9 | 62.3 | 79.5 | 66.5 | 52.0 | 60.1 |
>
> **W5.**
> >VQA and visual grounding task setup
>
> Thank you for pointing this out. We agree that text generation for VQA and object localization for visual grounding are more natural setups for these tasks. A ranking-based formulation may not be ideal for tasks where the target-side data is complex, such as the long-form answers in ScienceQA.
>
> Our motivation for including VQA and VG tasks in MMEB is to test models’ ability to represent concepts and objects as embedding vectors. We argue that the datasets selected in MMEB are well-suited for this purpose. Specifically, for VQA, we focus on answering tasks involving various visual concepts, supported by the observation that most answers are short-form (with an average length of fewer than 5 words). The objective is to challenge models to represent complex concepts across different domains and contexts.
> For VG, we evaluate models’ ability to represent the same object from different perspectives. In this setup, the query consists of the original image paired with an instruction pointing to the object of interest, while the target comprises a cropped version of the object, along with several distractor candidates.
>
> For more details and examples of VQA and VG tasks included in MMEB, please refer to Table 7 and 9 in our appendix.

---

> ### Author Response · Authors · 2024-11-27
> **Rebuttal followup**
>
> Dear Reviewer kzgZ,
>
> We would like to learn if our response addresses your concerns and questions, and we invite any additional feedback or thoughts for improving our paper. If you feel that our responses resolve the issues raised, we would be grateful if you could consider reflecting this in the evaluation. We would be happy to address any further concerns or questions. Thank you again for your time and effort!

---

> > ### Author Response · Authors · 2024-11-30
> > **Rebuttal followup**
> >
> > Dear Reviewer kzgZ
> >
> > Happy Thanksgiving! We really appreciate your feedback on our paper. We would like to learn if our response addresses your concerns and questions, and we invite any additional feedback or thoughts for improving our paper.

---

> > > ### Author Response · Authors · 2024-11-30
> > > **Rebuttal followup**
> > >
> > > Dear kzgZ
> > >
> > > As the discussion phase draws to the end, we kindly invite you to follow up on the discussion, and we would be happy to address any additional concerns or questions you might have.

---

> > > > ### Author Response · Authors · 2024-12-01
> > > >
> > > > Dear Reviewer kzgZ
> > > >
> > > > As the discussion phase draws to the end, we kindly invite you to follow up on the discussion, and we would be happy to address any additional concerns or questions you might have.

---

> > > > > ### Author Response · Authors · 2024-12-02
> > > > > **Final Day Reminder and Hope for Your Reply**
> > > > >
> > > > > Dear Reviewer kzgZ,
> > > > >
> > > > > This is a kind reminder that today is the final day of the discussion period. We greatly appreciate your time, efforts, and suggestions for our paper. Please let us know if our responses have addressed your concerns and questions.

---

### Official Review · Reviewer_bv32 · 2024-11-06

**Soundness:** 3
**Presentation:** 4
**Contribution:** 3
**Rating:** 8
**Confidence:** 4

**Summary:**

This paper presents to learn universal multimodal embedding models and introduces a novel benchmark (massive multimodal embedding benchmark, MMEB) to evaluate multimodal embedding.
The proposed VLM2VEC framework makes any state-of-the-art vision-language models (VLMs) multimodal embedding models through contrastive learning.
The MMEB dataset is collected from 36 public datasets (e.g. ImageNet, MSCOCO), covering 4 meta-tasks (e.g. VQA, retrieval).
Extensive experiments provide several baselines for multimodal embeddings and validate the effectiveness of the proposed VLM2VEC framework.

**Strengths:**

**[S1]** This paper is well-motivated and well-written.

**[S2]** I strongly agree with the requirement for a benchmark for evaluating multimodal embedding. The MMEB benchmark covers a wide range of tasks and is therefore sufficient to assess multimodal embeddings' capability appropriately.

**[S3]** This paper presents extensive experimental results with several baselines for multimodal embedding, providing a good benchmark for the following research.

**Weaknesses:**

**[W1] Novelty of VLM2VEC.**
The main concern is the limited novelty of the proposed VLM2VEC framework.
The proposed framework is significantly similar to the recent works that ground LLMs to visual data with pretrained image models and large language models, such as [1, 2].
In other words, projecting visual representations into LLM's input space and tuning the whole or partial model with text generation or contrastive objectives is a typical approach for the recent multimodal large language models.
Although I acknowledge the contribution to the purpose and necessity of this work, it would be good to emphasize the difference that the proposed framework is exclusive only for multimodal embedding.

```
[1] Li et al., "Visual instruction tuning," NeurIPS, 2024
[2] Xue et al., "xGen-MM (BLIP-3): A family of open large multimodal models," arXiv, 2024
```

**Questions:**

**[Q1] About [EOS] token in the learning objective.** VLM2VEC applies the InfoNCE loss to [EOS] tokens. I understand that the output [EOS] token reflects information gathered through the model’s attention over all preceding input tokens. Could the [EOS] token be replaced by a class token or learnable token? I expect that there will not be a big difference in performance. I wonder if the authors have a valid reason for taking [EOS] tokens or can provide results regarding output token selection.

**[Q2] Inference procedure.** [EOS] token of the output may be used as a single multimodal embedding. Although the inference procedure for each meta-task is well known, it would be good to explain the inference procedure through multimodal embedding. Specifically, I think that classification, VQA, and retrieval can be performed by measuring the similarity between outputs. However, it is not easy to guess how the inference of visual grounding is done.

---

> ### Author Response · Authors · 2024-11-25
> **Author response**
>
> Thank you so much for your constructive feedback to our work! The following is our response:
>
> **W1 Novelty of VLM2VEC**
>
> Thank you for your thoughtful feedback and for raising concerns about the novelty of VLM2Vec. We fully agree with your suggestion to emphasize the uniqueness of VLM2Vec as a framework exclusively designed for multimodal embedding tasks. Unlike prior works, VLM2Vec is specifically optimized for embedding tasks, such as retrieval, classification, and clustering, with a focus on precise cross-modal alignment and understanding. The novelty of this study can be summarized as follows:
> 1. MMEB offers a platform to showcase the capability of instructional multi-modal embedding capabilities, systematically testing the effectiveness of embedding models. VLM2Vec demonstrates strong meta-task generalization capabilities, as shown through its evaluation on MMEB—a unified benchmark designed to systematically test embedding models across modalities and diverse tasks.
> 2. Additionally, VLM2Vec provides a scalable framework for adapting any VLM to an embedding model, enabling seamless integration of pretrained vision models and LLMs for embedding purposes. As demonstrated in Section 4.4, scaling experiments reveal that further scaling significantly improves model performance, and we are far from reaching the limits of this trend. Building on this, we intend to expand the VLM2Vec family by utilizing different backbone models and exploring various scaling factors. Our future work aims to release more models in the VLM2Vec family to further establish the generalizability and flexibility of this framework.
>
> In summary, VLM2Vec represents a unique contribution as a dedicated multimodal embedding framework, which can organically complement with regular generative approaches on tasks requiring accessing large volume of data or very long context understanding. We will revise the manuscript accordingly to better articulate these distinctions. Thank you again for your valuable suggestions!
>
> **Q1: About [EOS] token in the learning objective**
>
> Thank you for your insightful question regarding the use of the [EOS] token for pooling in VLM2Vec. We chose the [EOS] vector for pooling based on findings from the text embedding work E5-Mistra[1] and NV-Embed [2]. As shown in E5-Mistral [1], using alternative pooling methods instead of the [EOS] token may result in small performance drops. However, we believe that extensive training could lead to a minimum gap between different pooling tokens. While we focused on [EOS] pooling in this study, we appreciate your suggestion and will explore the impact of other pooling methods, such as class tokens or learnable tokens in the revision.
> [1] E5-mistral: Wang, Liang, et al. "Improving text embeddings with large language models." arXiv preprint arXiv:2401.00368 (2023).
> [2] NV-Embed: Lee, Chankyu, et al. "NV-Embed: Improved Techniques for Training LLMs as Generalist Embedding Models." arXiv preprint arXiv:2405.17428 (2024).
>
> **Q2: Inference procedure for visual grounding**
>
> Thank you for raising this important question regarding the inference procedure for different tasks. For visual grounding (VG), the inference procedure is also based on similarity prediction with task-specific instructions. Specifically:
> Input Representation: The query combines an instruction (e.g., "Select the portion of the image that isolates the object of the given label: red apple") with the full image. This instruction guides the model to focus on a specific object within the image.
> Target Representation: Each target corresponds to cropped regions (bounding boxes) of the image, including both the object of interest and distractor regions.
> Embedding Similarity: The multimodal embedding of the query ([EOS] token representing the full image and instruction) is compared to the embeddings of each cropped region using cosine similarity. The cropped region with the highest similarity score is selected as the predicted grounding location. This approach effectively uses instructions to represent the same object from two perspectives: the full image and its specific cropped regions.
> You can refer to the visual grounding data examples provided in the appendix (Page 24, Table 9) for additional clarity. Thank you for highlighting this point; we will ensure the revised manuscript includes a clear explanation of the inference procedures. We truly appreciate your valuable feedback!

---

> > ### Comment · Reviewer_bv32 · 2024-11-30
> >
> > Thank you for your response. I still think this work contributes to multimodal learning research. I would like to keep my initial rating.

---

> > > ### Author Response · Authors · 2024-11-30
> > >
> > > Thank you so much for maintaining your positive assessment of our work! We sincerely express our gratitude for your constructive comments and suggestions!

---

### Author Response · Authors · 2024-11-26
**Summary of Revision**

We thank the reviewers for the time and expertise they devoted to reviewing our paper. We appreciate that the reviewers found our work to be well-motivated, recognized the value of our proposed MMEB benchmark, and acknowledged that our extensive experiments and ablations demonstrate the effectiveness of the VLM2Vec framework.

And we are grateful for the constructive feedback. To help better address the questions and concerns raised by the reviewers and further improve the paper's quality, we made the following updates to the paper PDF. These contents are also used in our response to the reviewers

* Benchmark design (**Reviewer kzgZ**): We have added **Appendix A.2** and **Table 5** to explain why we chose 1,000 as the number of candidates, striking a balance between evaluation cost and benchmark difficulty.

* Baseline evaluation using the best version of the UniIR model (**Reviewer kzgZ**): We have updated the results of this baseline model in our **Table 2**.

* Baseline CLIP-based model with instruction (**Reviewer hG4f**): We have updated **Section 4.3.4** and **Table 4** to discuss the impact of instructions on the CLIP-based model, explain why we do not use instructions with the CLIP-based model, and highlight VLM2Vec’s instruction-following capability.

* Fine-tune baseline models on MMEB training sets and include the training setup (Reviewer **kzgZ**, Reviewer **hG4f**): We have updated **Section 4.1** and **Table 2** to include two baseline models fine-tuned on our own datasets.

* More clarifications for Figure 5 (**Reviewer hG4f**): We have updated the caption of **Figure 5** for improved clarity.

* On OOD Evaluation (**Reviewer hG4f**): We have added the results of LlaVA-1.6-based VLM2Vec to **Table 2**. Compared to Phi-3.5-v, LlaVA-1.6 offers a transparent training recipe, and the OOD datasets from our MMEB are valid. We have demonstrated that VLM2Vec is an effective framework capable of supporting different VLMs.

* On Evaluating Traditional Benchmarks (**Reviewer hG4f**): We have added zero-shot retrieval performance on Flickr30K in **Table 11**.

* On inference procedure for visual grounding (Reviewer **bv32**): We have updated **Section 2.2** to clarify the description of the visual grounding task.

Once again, we would like to thank all the reviewers for their constructive suggestions, which have helped us improve our work.

---

### Meta-Review · Area_Chair_HBm9 · 2024-12-21

**Metareview:**

This paper studies two tasks: 1. evaluate the multimodel embedding. 2. how to convert the VLM to a embedding model. For the task 1, this paper proposed a massive benchmark: MMEB, which can comprehensively evaluate the embedding model performance. For the task 2, the paper proposed VLM2Vec, which is LoRA finetuned VLM. The model is optimized via contrastive learning.

Strength:
1. The multimodal embedding evaluation task is a quite important problem, while it has been fully explored compared to the text side.
2. The VLM2Vec is a great idea for converting the VLM to embedding model. The LoRA finetuned version shows great potential.

Weakness:
1. The reviewer argues the fairness (# of parameters) in the comparison b/w CLIP and the VLM
2. The reviewer argues the VLM might train on additional data which might achieve advantage on those benchmark over CLIP.
3. The negative space of the benchmark might not enough.

After reading the paper and the rebuttal, based on my judgement and my experience in multimodal LLM and representation learning, I think it is important to have a massive benchmark can reflect the model performance, as imagenet and coco retrieval is hard to justify the effectiveness of the approach. Furthermore, how to convert the VLM to Vec is an open question in both academia and industry. I think this paper proposed a simple yet effective approach to tackle this open question.

Given the above consideration, I would recommend Accept.

**Additional Comments On Reviewer Discussion:**

The review mainly argues the parameter and the addtional data used in the VLM training might bring advantages over the CLIP (traditional approach) There is a debating b/w the reviewer hG4f and the author. After reading the rebuttal, I would lean towards the author.

It is important to improve the performance envelope of the embedding model. VLM shows great potential and performance in multiple tasks, however it is still challenging to use it for representation / embedding. I think VLM2Vec is a interesting yet effective way to achieve this.

Fairness in training data / training data leakage: fairness is important. However it is hard to trace all the training data even for CLIP model. It is hard to justify that the VLM might have an edge of using data might have overlap with the evaluation benchmark.

Based on this thinking, I think the author fully answered the reviewer's concerns.

---

### Decision · Program_Chairs · 2025-01-22

Accept (Poster)